# Spatiotemporal variations in atmospheric CH₄ concentrations and enhancements in northern China based on a comprehensive dataset: Ground-based observations, TROPOMI data, inventory data and inversions

Pengfei Han[1,2*], Ning Zeng[3], Bo Yao[4], Wen Zhang[5], Weijun Quan[6], Pucai Wang[2], Ting Wang[2], Minqiang Zhou[1,2], Qixiang Cai[7,8*], Yuzhong Zhang[9,10], Ruosi Liang[9,10], Wanqi Sun[11*], Shengxiang Liu[2,12]

[1]Key Laboratory of Atmospheric Environment and Extreme Meteorology, Institute of Atmospheric Physics, Chinese Academy of Sciences, Beijing 100029, China

[2]Carbon Neutrality Research Center, Institute of Atmospheric Physics, Chinese Academy of Sciences, Beijing, China

[3]Department of Atmospheric and Oceanic Science, and Earth System Science Interdisciplinary Center, University of Maryland, College Park, Maryland, USA

[4]Department of Atmospheric and Oceanic Sciences and Institute of Atmospheric Sciences, Fudan University, Shanghai, China

[5]State Key Laboratory of Atmospheric Boundary Layer Physics and Atmospheric Chemistry, Institute of Atmospheric Physics, Chinese Academy of Sciences, Beijing, China

[6]Institute of Urban Meteorology (Key Laboratory of Urban Meteorology), China Meteorological Administration, Beijing, China

[7]State Key Laboratory of Numerical Modeling for Atmospheric Sciences and Geophysical Fluid Dynamics, Institute of Atmospheric Physics, Chinese Academy of Sciences, Beijing, China

[8]Qiluzhongke Institute of Carbon Neutrality, Jinan, China

[9]Key Laboratory of Coastal Environment and Resources of Zhejiang Province, School of Engineering, Westlake University, Hangzhou, Zhejiang, China

[10]Institute of Advanced Technology, Westlake Institute for Advanced Study, Hangzhou, Zhejiang, China

[11]Meteorological Observation Centre, China Meteorological Administration, Beijing, China

[12]Jiujiang University, Jiujiang, Jiangxi, China

*  *Correspondence to*: Pengfei Han (pfhan@mail.iap.ac.cn); Qixiang Cai (caiqixiang@mail.iap.ac.cn); Wanqi Sun (sunwanqi2008@126.com)

**Abstract.** Methane ($CH_4$) is a potent greenhouse gas with a global warming potential that is 28–36-fold higher than that of $CO_2$ at the 100-year scale. Northern China notably contributes to $CH_4$ emissions. However, high uncertainties remain in emissions, and observation gaps exist in this region, especially in urban areas. Here, we compiled a comprehensive dataset (available at https://doi.org/10.5281/zenodo.10957950) (Han et al., 2024), including ground- and satellite-based observations, inventory data and modeling results, to study the $CH_4$ concentration, enhancement and spatiotemporal variation in this area. High-precision in situ observations from Beijing and Xianghe revealed that obvious seasonal cycles and notable enhancements (500–1500 ppb) occurred at a regional background site (Shangdianzi). We found significant increasing trends in the $CH_4$ concentration over time in both the ground- and satellite-based observations and positive correlations between these observations. Anthropogenic emissions largely contributed to surface concentration variations and their increases in middle and southern Shanxi Province and Hebei Province. The posterior concentrations generally agreed well with the surface in situ observations (mean biases ranging from -2.3~80.7 ppb), and the RMSE error ranges from 110 to 185 ppb, which is in the range from 5% to 10% of the $XCH_4$. Moreover, a generally spatially consistent pattern was observed between the results of posterior results and the Tropospheric Monitoring Instrument (TROPOMI) column $CH_4$ observations in four seasons. The posterior surface $CH_4$ concentrations (with a spatial resolution of $0.5° \times 0.625°$) revealed that southern Shanxi, northern Henan, and Beijing exhibited relatively high levels (an increase of ~300 ppb), which were positively correlated with the PKU-$CH_4$-v2 emission inventory data. The inversion results using TROPOMI observations was 24.0 Tg, a decrease of 15.6%, or 4.4 Tg, compared with the prior EDGARv4.3.2 (28.5 Tg). This study provides a comprehensive data set of $CH_4$ concentrations and enhancements in high-emission areas, which can benefit the research community and policy-makers for designing future observations, conducting atmospheric inversions and formulating policies, and city level with high spatial-resolution at km scale atmospheric inversions are highly needed.

Keywords: Methane, in situ measurements, TROPOMI, TCCON, emissions inventory; atmospheric inversions

# 1    Introduction

Methane ($CH_4$) is a potent greenhouse gas (GHG) that exhibits a 28–36-fold greater global warming potential than that of $CO_2$ at the 100-year scale (Hu et al., 2024; Lin et al., 2021), with a radiative forcing of 0.61 W m$^{-2}$, and $CH_4$ is responsible for almost one-third of the total warming to date (Etminan et al., 2016; IPCC, 2022). According to National Oceanic and Atmospheric Administration (NOAA) atmospheric observations, the global mean atmospheric $CH_4$ growth rate increased dramatically to 13.2 ppb in 2022, resulting in record-high $CH_4$ levels above 1900 ppb throughout 2022 (https://gml.noaa.gov/ccgg/trends_ch4/). The fluctuations in the atmospheric $CH_4$ concentration are driven by various natural (e.g., wetlands) and anthropogenic sources (e.g., fossil fuel exploitation), and atmospheric $CH_4$ can be removed by sinks via chemical oxidation involving hydroxyl radicals (OH) and dry soil sinks involving aerobic methane-oxidizing bacteria (Lin et al., 2021; Saunois et al., 2020; Tan et al., 2022; Turner et al., 2019). Anthropogenic sources contribute approximately 60% to global $CH_4$ emissions (Jackson et al., 2020; Saunois et al., 2020). Thus, reductions in anthropogenic $CH_4$ emissions have significant implications for achieving near-term climate goals (Gouw et al., 2020; IPCC, 2022; Staniaszek et al., 2022). To limit global warming to 1.5°C, more than 130 countries have pledged to achieve carbon neutrality or net-zero emissions, which requires the combined reduction in both $CO_2$ and non-$CO_2$ (GHG) emissions (Fankhauser et al., 2022; Ou et al., 2021).

Direct emissions (e.g. leakage/non-fully combustion from energy storage, transportation and consumption, landfills and waste water) originating from urban areas account for approximately 21% of global $CH_4$ emissions (Zhao et al., 2019). For example, Crippa et al. (2021) reported that urbanization contributed to a six-fold faster increase in $CH_4$ emissions stemming from urban centers and that energy, transport, and waste were the dominant drivers of increases in urban emissions. Since 2000, $CH_4$ emissions in China have rapidly increased in response to industrialization and urbanization development (Lin et al., 2021). Accompanying this trend, notable expanding hotspots in mega-cities and high-energy-exploitation regions have become a concern. China enacted an ambitious plan to reach carbon neutrality before 2060 to address climate change. In November 2023, China issued the Methane Emissions Control Action Plan, which targets a utilization volume of coal mine methane of 6 billion cubic meters, a utilization rate of urban household waste of approximately 60%, and a utilization rate of dung and waste from livestock of at least 80% by 2025 (MEE, 2023). Understanding the current

emission status, impacts on atmospheric $CH_4$ concentration increases, and mitigation potentials for $CH_4$ emissions are prerequisites for developing effective mitigation policies.

Although previous efforts have been made to improve the accuracy of $CH_4$ emission estimates for China, substantial inconsistencies remain, especially in hotspot regions (Lin et al., 2021; Liu et al., 2021b; Miller et al., 2019; Sheng et al., 2019b). The recent emission inventories of PKU-$CH_4$ v2 (Liu et al., 2021b), Community Emissions Data System (CEDS) v2021-4-21 (Hoesly, 2019) and Emissions Database for Global Atmospheric Research (EDGAR) v7.0 (Crippa, 2023) exhibit a wide range of

47–67 Tg for 2019, which highlights the considerable uncertainty in the application of bottom-up methods. These uncertainties are mainly due to differences in source-specific emission factors and spatial disaggregation of national or provincial annual totals (Crippa, 2023; Lin et al., 2021; Peng et al., 2016; Zhang et al., 2016). Furthermore, differences among inventories could substantially affect inversions using inventory data as prior estimates. The adoption of data from existing top-down studies

(Miller et al., 2019; Yin et al., 2021) based on outdated bottom-up inventories could bias the determination of trends in $CH_4$ emissions in China (Liu et al., 2021b). Tan et al. (2022) also reported that the inversion model performance is highly affected by prior data and measurements across China. There is a pressing need to improve the accuracy of $CH_4$ emission estimates to support the implementation of mitigation strategies and better characterize regional $CH_4$ surface fluxes.

Satellite observational platforms provide promising pathways for tracking spatial and temporal variations in $CH_4$ sources (Irakulis-Loitxate et al., 2021; Jacob et al., 2016; Pandey et al., 2019; Schuit et al., 2023; Turner et al., 2015). Satellite retrievals of the column-averaged dry air mole fraction of methane (X$CH_4$) with an unprecedented spatiotemporal coverage and resolution can be used to rapidly detect $CH_4$ variations and verify bottom-up inventories. Although several previous studies have

involved the use of data from the Greenhouse Gases Observing Satellite (GOSAT) and the SCanning Imaging Absorption Spectrometer for Atmospheric Chemistry (SCIAMACHY) to characterize atmospheric $CH_4$ concentrations in China, the monitoring of emissions originating from large sources remains limited because of the relatively sparse observations and coarse resolution (Chen et al., 2022a; Chen et al., 2022b; Tan et al., 2022). Furthermore, Plant et al. (2022), Maasakkers et al. (2022), and

Peng et al. (2023) reported that inventoried urban $CH_4$ emissions are underestimated relative to Tropospheric Monitoring Instrument (TROPOMI)-based estimates. Several studies have shown the

ability of the recently launched TROPOMI to track and quantify $CH_4$ emissions stemming from point and regional sources (Barré et al., 2021; Jacob et al., 2016; Schuit et al., 2023). Gouw et al. (2020) reported that the TROPOMI can identify distinct methane emission increases in oil and natural production regions in the United States. Liu et al. (2021c) developed a new divergence method to estimate $CH_4$ emissions in Texas (North America) on the basis of TROPOMI observations. Liang et al. (2023) used TROPOMI observations to estimate emissions in East Asia.

Northern China, encompassing the Beijing–Tianjin–Hebei (BTH) region and its surrounding provinces (including Shanxi, Shandong, Jiangsu, Anhui, and Henan), is a populous region with rapid socioeconomic development, and more than 30% of the anthropogenic $CH_4$ emissions in China in 2019 was generated in this region (PKU-$CH_4$, Fig. 1). Previous studies have indicated that northern China is a $CH_4$ emission hotspot region (Liang et al., 2023; Tan et al., 2022). Emissions resulting from the production of raw coal in northern China constitute one of the major sources, and Shanxi is the largest regional $CH_4$ emitter, yielding 5.7 Tg of emissions in 2019 (PKU-$CH_4$). Notably, northern China is a hotspot region for atmospheric $CH_4$ concentration and flux studies.

In this study, we used high-precision in situ observations, Total Carbon Column Observing Network (TCCON) observations, satellite data, inventory data, and modeling data from atmospheric inversions to better understand the spatiotemporal variations and spatial gradients of atmospheric $CH_4$ concentrations and the correlations between emissions and concentrations in northern China. On the basis of this comprehensive dataset, we aimed to (1) quantify the spatiotemporal variation of $CH_4$ concentrations and enhancements in northern China; (2) study the correlations between satellite- and ground-based observations; and (3) assess the consistency and deviation in results derived from surface and satellite observations. First, we studied the temporal variations in local $CH_4$ concentrations and their enhancement in urban areas. Second, we analyzed the correlations between satellite-based column $CH_4$ concentrations and surface observations. Third, we assessed the model performance via high-precision measurements. Finally, we analyzed the spatial and temporal variations in posterior concentrations determined with the Westlake model, which exhibits a satisfactory output and performance, at the monthly, seasonal, and yearly scales.

## 2 Data and methods

 ## 2.1 Surface observations

To monitor GHG emissions in support of assessing the realization of carbon neutrality goals, China is making great efforts in terms of its GHG monitoring capacity (Han et al., 2018; MEE, 2021; Sun et al., 2022; Zeng et al., 2021). Three stations equipped with high-precision (1 ppb) Picarro instruments have been established in the BTH region, namely, the urban Beijing station (BJ), the suburban Xianghe station (XH), and the WMO/GAW regional background Shangdianzi station (SDZ) (Fig. 1, Table 1), since 2019. Wind rose plots for data from these sites are shown in Fig. S1. Moreover, two TCCON stations (http://www.tccon.caltech.edu/), namely, the Hefei and Xianghe stations, were established to continuously monitor the variability in the atmospheric $XCH_4$. In this study, we analyzed surface measurements along with satellite observations to better understand the temporal variations and seasonal cycles of atmospheric $CH_4$ from 2019 to 2021, while TCCON data were also employed to assess TROPOMI observations.

Table 1 Information on the three regional high-precision observation sites

| Station name | Abbreviations | Station type | Longitude (°E) | Latitude (°N) | Altitude (m) | Height of the inlet (m) |
|---|---|---|---|---|---|---|
| Beijing | BJ | Urban | 116.3667 | 39.9667 | 49 | 80/280 |
| Xianghe | XH | Suburban | 116.9578 | 39.7833 | 95 | 60/100 |
| Shangdianzi | SDZ | Regional background | 117.1166 | 40.6500 | 293 | 16/80 |

2.1.1 Ground-based high-precision in situ measurements

In situ measurements of atmospheric $CH_4$ dry mole fractions were conducted at the three sites (BJ, XH, and SDZ) via Picarro GHG analyzers (Fig. 1). The BJ station (116.37°E, 39.97°N) is located at the Institute of Atmospheric Physics, Chinese Academy of Sciences, in urban Beijing between the Third and Fourth Ring Roads (Liu et al., 2021a). This area is densely populated, and $CH_4$ concentrations are frequently influenced by local residential and transportation emissions. The XH station (39.75°N, 116.96°E) is located at a suburban site that represents the transition region from urban to regional

background areas (Yang et al., 2021). The SDZ station (117.12°E, 40.65°N) is one of the regional Global Atmosphere Watch (GAW) stations of the World Meteorological Organization (WMO) in China and occurs on a mountainside 100 km northeast of urban Beijing. There is a small village in the lower

valley of the mountain. The major vegetation types are shrubs and corn (Fang et al., 2016). The Mona Loa (MLO, 19.54°N, 155.58°W) site is a GAW station representing the global background (not shown in Fig. 1) located atop a mountain on Hawaii Island with the longest history of observations. The background map in Fig. 1 shows 10 km×10 km gridded anthropogenic $CH_4$ emissions from the PKU-$CH_4$ inventory with hotspots in Shanxi, Beijing, Henan, Anhui, and Inner Mongolia, while the

subplots show sectoral $CH_4$ emissions from 2000 to 2019 at the provincial scale (Figs. 1 and S2).

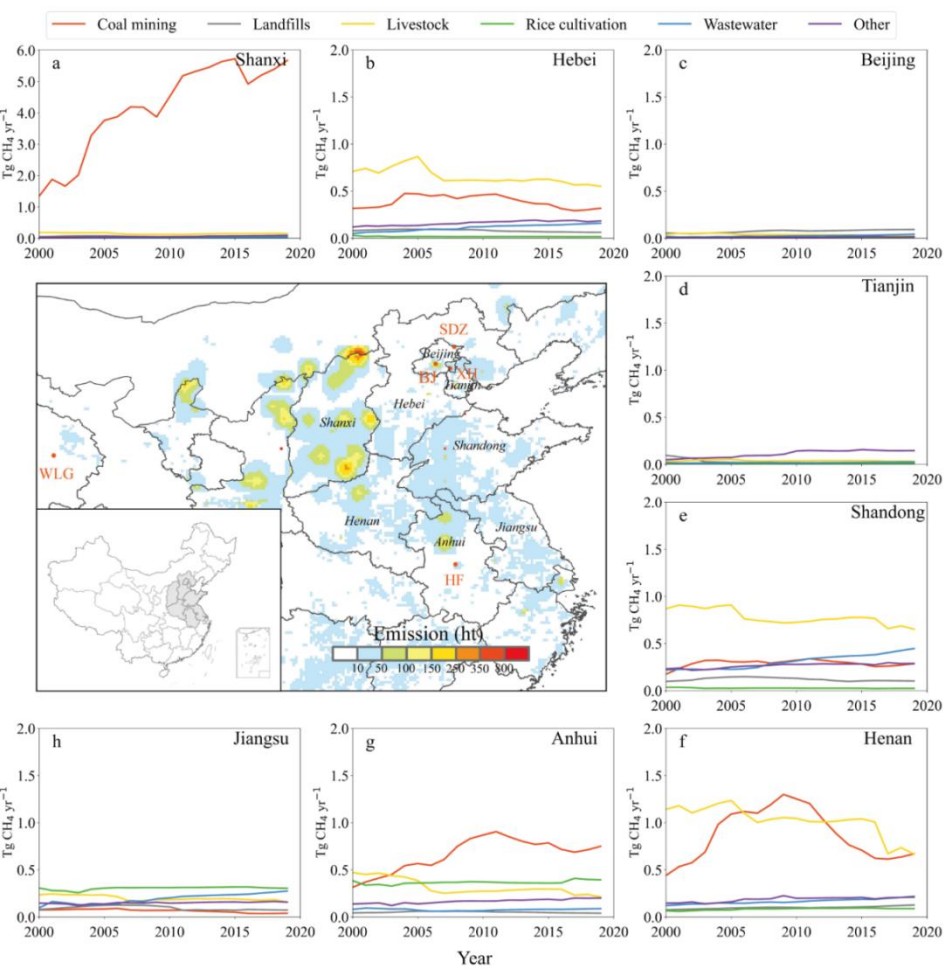

**Fig. 1 Observation sites (red dots) and gridded anthropogenic CH₄ emissions and sectoral emissions in northern China from the PKU-CH₄-v2 inventory (Peng et al., 2024). BJ, XH, SDZ, and HF represent Beijing urban, Xianghe suburban, Shangdianzi regional background, and Hefei suburban conditions, respectively**
**(Table 1). Methane emissions from coal mining, landfills, livestock, rice cultivation, wastewater, and others were presented in the subplots for each province. The unit ht and Tg denote hundred tons and Tera-gram ($10^{12}$).**

Cavity ring-down spectroscopy (CRDS) instruments (Picarro G2301/G2401, Picarro Inc.) were used to continuously measure in situ atmospheric $CH_4$ concentrations at BJ, XH and SDZ. $CH_4$ concentration data from the Waliguan (WLG) and MLO sites were obtained from the World Data Center for Greenhouse Gases (WDCGG; https://gaw.kishou.go.jp/). At the three regional sites, ambient air was sampled by an oil-free vacuum pump at different tower levels after particles were removed with a 2-μm filter. Then, the air was dried with a Nafion dryer to the dewpoint at –25°C, and the pressure and flow rate were stabilized before the air samples were analyzed with a CRDS instrument. All the observation systems were calibrated every 6 hours by WMO X2007 standard gases. The accuracy of the observations was greater than 1 ppb at a 1-minute resolution. The sampling height was 80/280 m above ground level at BJ, 60/100 m at XH, and 16/80 m at SDZ.

2.1.2 Ground-based total column measurements

The TCCON aims to measure column-average mole fractions of $CO_2$, $CH_4$, and other gases beginning in 2004 across 30 sites worldwide via solar absorption spectroscopy in the near-infrared region (Laughner et al., 2023). It is a ground-based network of Fourier transform spectrometers (FTSs) designed to retrieve high-precision data on GHG emissions and to provide a validation dataset for space-based measurements (Wunch et al., 2011; Wunch et al., 2010). Here, we used GGG2020 TCCON data from the Hefei (HF, 117.17°E, 31.9°N) and XH stations (Liu, 2023; Zhou, 2022). A high-resolution FTS (IFS125HR, Bruker GmbH, Germany) system and a solar tracker (Tracker-A Solar 547, Bruker GmbH, Ettlingen, Germany) have been installed at the HF site since January 2014 (the site location shown in Fig. 1) (Wang et al., 2017). The observatory is located in the northwestern suburbs of Hefei and is surrounded by wetlands and croplands (Tian et al., 2018). Therefore, the $CH_4$ concentration observed at the HF station may be partly influenced by local anthropogenic emissions from urban areas and cultivated lands and by natural emissions from wetlands (Tian et al., 2018; Wang et al., 2017). The bias correction factor for $CH_4$ at HF is 0.9765, with a 1σ standard deviation of 0.0020 (Tian et al., 2018). Additionally, an automatic weather station (ZENO 3200, Coastal Environmental Systems, Inc., Seattle, USA) was installed near the solar tracker instrument on the roof in September 2015 to collect meteorological data (Shan et al., 2019; Wang et al., 2017). The other TCCON site is located at XH in the suburban area 50 km southeast of Beijing (Zhou et al., 2023). The XH station is

surrounded by croplands and residential buildings (with an average height of ~20 m) (Yang et al., 2021). A Bruker IFS 125HR instrument was installed in the upper level of a four-story building in June 2016, and a solar tracker instrument was installed on the rooftop in June 2018 (Yang et al., 2021). The retrieved XCH$_4$ products at XH are subjected to air-mass-dependence correction and calibrated to the WMO scale (Wunch, 2015; Yang et al., 2020). Moreover, the Xianghe and Hefei sites have percentages of 32.6% and 87.1% in 2019, respectively, for high AOD (polluted or cloudy) that do not retrieve CH$_4$. With high aerosol or cloud conditions, the DC signal of the interferogram has a large variation. All the spectra with the DC variation larger than 5% are filtered out, which guarantees the data quality of the TCCON spectra. In addition, the TCCON uses the solar direct absorption spectra which has a much less impact from the low aerosol as compared to the satellite retrieval.

## 2.2 Satellite observations

The TROPOMI onboard the Copernicus Sentinel-5 Precursor is a nadir-viewing, imaging spectrometer covering wavelength bands between the ultraviolet and shortwave infrared (SWIR) bands (Veefkind et al., 2012). The TROPOMI retrieves a methane column from the 2305–2385-nm SWIR band and the 757–774-nm near-infrared band, with a daily global coverage at a fine spatial resolution of 5.5 km×7 km since August 2019 (7 km×7 km before August 2019) and a swath width of ~2600 km (Butz et al., 2012; Hu et al., 2016; Lorente et al., 2021). We used the TROPOMI CH$_4$ total column level 2 data product to quantify the variations and trends in northern China from January 2019 to December 2021. We employed XCH$_4$ retrievals with quality values greater than 0.5 (Gouw et al., 2020). To ensure comparison with surface measurements, bottom-up inventories and inversion results at different spatial resolutions, the TROPOMI XCH$_4$ observations were averaged to three spatial resolutions of 0.1°×0.1°, 0.25°×0.25°, and 0.5°×0.5°. The TROPOMI data were resampled to each of the spatial resolutions. We first defined the spatial bounds of the resampled grids and then placed the original pixels into coarser grids according to the longitude and latitude of the pixel center. We defined the resampled values of XCH$_4$ as the average of the original pixels belonging to the new resampled grids. This method ensures consistency between the regionally averaged XCH$_4$ values before and after resampling, with an average relative error of 0.03%. We used TCCON data from HF and XH to evaluate the accuracy and precision of the TROPOMI observations.

**2.3 Bottom-up inventory**

A gridded inventory of anthropogenic $CH_4$ emissions from Peking University (PKU-$CH_4$ v2) (Liu et al., 2021b; Peng et al., 2023; Peng et al., 2022), which has been assessed in our previous study (Lin et al., 2021), was adopted in this study. PKU-$CH_4$ v2 is an annual bottom-up inventory based on provincial activity data and regional, sector-specific emission factors for eight major sectors in China (Liu et al.,

2021b; Peng et al., 2016). The inventory provides a priori knowledge of the temporal and regional distribution characteristics of anthropogenic $CH_4$ emissions in China. The main sources of $CH_4$ emissions in China are coal mining and agriculture, which contributed approximately 77% to the total national emissions in 2019 (Lin et al., 2021; Liu et al., 2021b).

Coal mining is the dominant driver of $CH_4$ emissions in China, accounting for >80% of the increase in

the total emissions in the 2000s due to the growth in coal production with rapid economic development and the increasing energy demand (Lin et al., 2021; Liu et al., 2021b). However, the reductions in both coal production and emission factors, with increasing utilization rates, contributed to slowing coal methane emissions from 2010–2019 (Liu et al., 2021b).

**2.4 Atmospheric modeling and inversions**

We present a Bayesian inversion framework over East Asia, using GEOS-Chem as the forward model, with a spatial resolution of $0.5° \times 0.625°$ (Liang et al., 2023). The state vector to be optimized in the inversion consists of 600 clusters for methane emissions and average methane column biases at four model boundaries. The inversion-derived posterior simulations (referred to as Westlake data), which provide an improved fit to the TROPOMI observations, are considered in our study (Liang et al., 2023).

For oil and gas, we used the Global Fuel Exploitation Inventory (GFEI) v1.0 dataset, and for coal in China, we used the inventory of Sheng et al. (2019b), while the data for other sectors were derived from the EDGAR v4.3.2 dataset (Janssens-Maenhout et al., 2019a). We used annual coal emissions (GFEI v1.0), and annual livestock emissions (annual EDGAR v4.3.2), which are evenly distributed across time, but used monthly rice emissions, which is annual EDGAR v4.3.2 scaled with seasonal

scaling factors (higher in autumn and lower in other seasons). To consider major patterns in the distribution of emissions and significantly reduce the inversion computation burden, emissions were optimized on the basis of 600 spatial clusters instead of the native $0.5° \times 0.625°$ grid, which were generated with a Gaussian mixed model algorithm (Turner and Jacob, 2015). The model performance

was evaluated by high-precision in situ observations. The optimized surface and column concentrations were used to analyze the spatiotemporal dynamics of $CH_4$ concentrations and emissions. Moreover, Copernicus Atmosphere Monitoring Service (CAMS) global inversion-optimized greenhouse gas concentrations were used in the comparison, which are coarse-resolution ($2° \times 3°$) monthly data that can provide a regional baseline (Rayner et al., 2016).

## 3 Results and discussion

### 3.1 Temporal variations in the in situ $CH_4$ concentrations in urban areas determined by ground-based high-precision measurements

To understand the errors in concentrations and emissions in the study area, we firstly analyzed the temporal variations and the spatial enhancements on the basis of ground-based observations. There were clear temporal variations in the high-precision in situ $CH_4$ concentrations (Fig. 2-3) at all three sites from the urban BJ station to the suburban XH station to the regional background SDZ station. The concentrations at BJ and XH ranged from ~2000 ppb for the baseline to 4000–5000 ppb for the peak values in September, and November to February in some heavily polluted cases in winter. We further plotted frequencies of higher than 2500ppb for each month (Fig. S5), and autumn and winter months reached a higher 20%+ than spring and summer for BJ. These results could be associated with high emissions from nearby wetlands at XH in summer (July) and high residential and natural gas power plant emissions at BJ in winter (Ji et al., 2020).

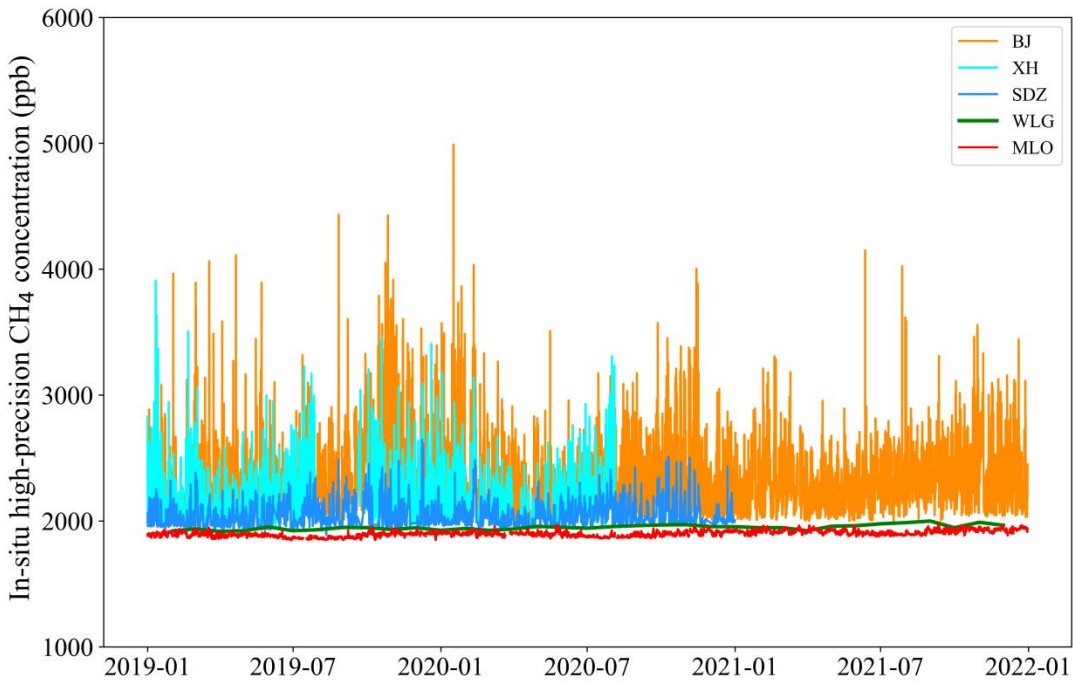

**Fig. 2 Hourly CH₄ variations at the four sites in northern China on the basis of ground high-precision observations. BJ, XH, SDZ, WLG, and MLO represent Beijing, Xianghe, Shangdianzi, Waliguan, and Mona Loa, respectively. For WLG, we used weekly data because of data availability.**

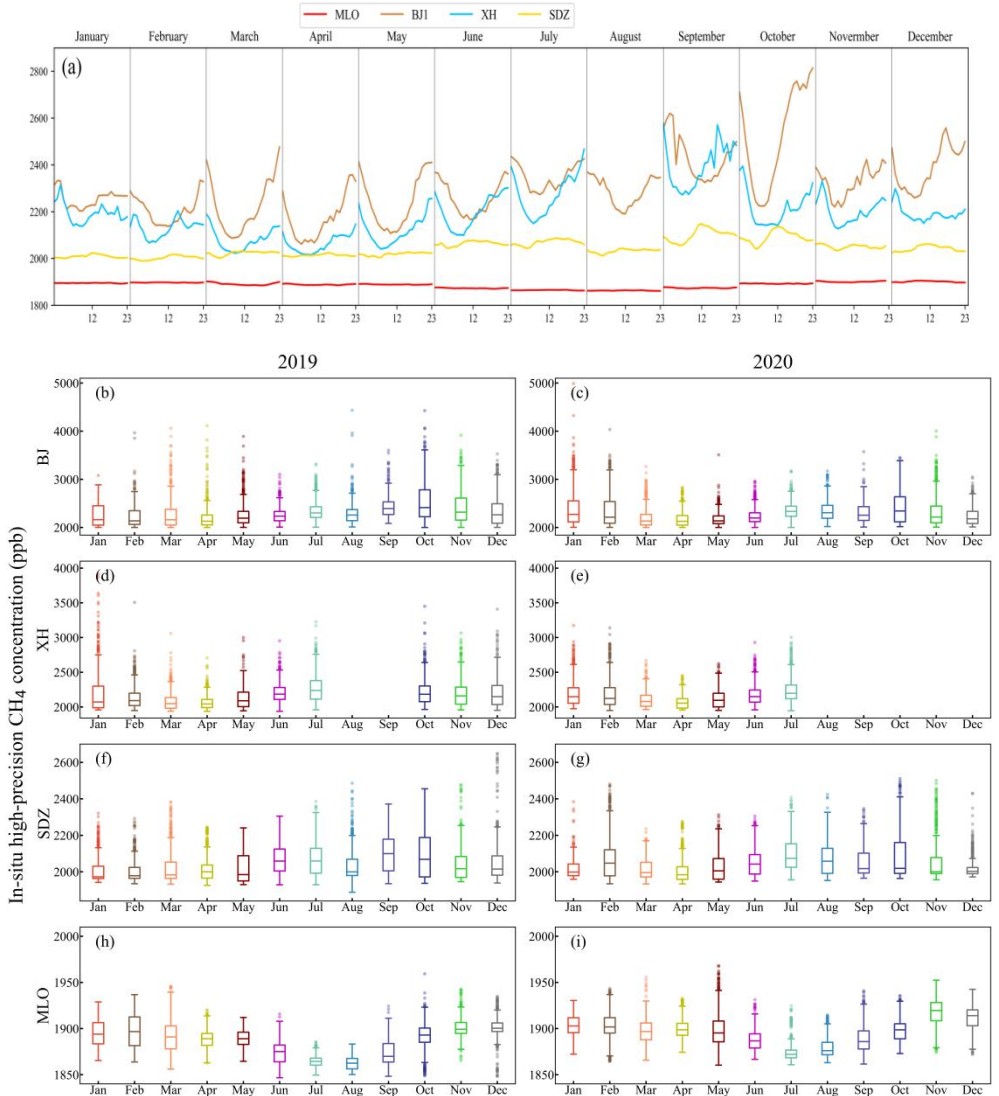

Fig. 3 CH$_4$ concentrations for the hourly mean from 00:00–23:59 (UTC time) in each month, and boxplots of monthly concentrations in the four sites in 2019 and 2020. BJ, XH, SDZ, and MLO represent Beijing, Xianghe, Shangdianzi, and Mona Loa, respectively.

We compared our results with other observations in this region and similar latitudes in the USA. The results were consistent with the hourly concentrations (2000-6000ppb) and enhancements (0-1800ppb) in Xiaodian, near Taiyuan, Shanxi (Hu et al., 2023). Such high concentrations have also been observed in large cities in Canada and the U.S., such as Los Angeles (LA) (Verhulst et al., 2017), Washington, D.C., and Baltimore (WA) (Huang et al., 2019), and Indianapolis (IN) (Mitchell et al., 2022) (Fig. 4). And for overall hourly means of the 2019, BJ showed the highest concentrations, followed by LA and XH, and almost all sites showed diurnal variations with a peak around 08:00 local time (Fig. 4), due to

305 the lower planetary boundary layer height (PBLH) and accumulation of emissions (Fig. S4-S5) like

$CO_2$ (Bao et al., 2020; Zeng et al., 2021) and air pollutants (Chu et al., 2019; Su et al., 2018). Monthly

mean data also showed high concentrations in autumn and winter in all sites, with BJ and XH much

stronger, although with large hourly and daily variations (Fig.3, Fig. S4-S5). Furthermore, the WLG

site (green line) provided a continental baseline, while the MLO site (red line) provided a global

baseline. There were also notable seasonal cycles and high spatial gradients at the three regional sites

(Figs. 2, 3 and 4, respectively). In contrast, SDZ and WLG exhibited fewer seasonal cycles (Fig. 2-4).

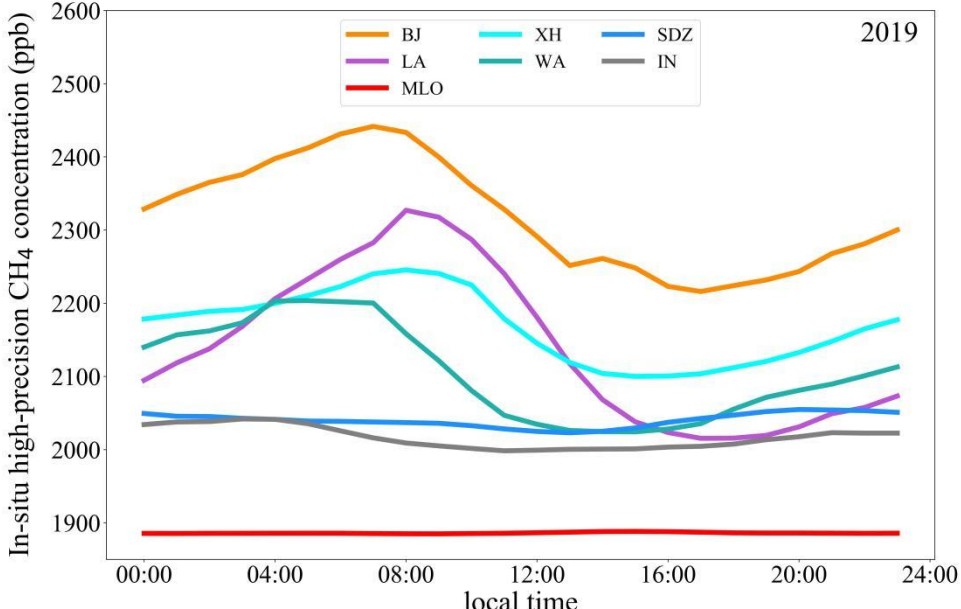

**Fig. 4 Hourly mean of CH₄ concentrations from 00:00－23:59 (local time) in 2019 at northern China and**

**comparison sites in the USA (Mitchell et al., 2022). BJ, XH, SDZ, LA, WA, IN, and MLO denote Beijing,**

**Xianghe, Shangdianzi, Los Angeles, Washington, Indianapolis, and Mona Loa.**

## 3.2 Temporal variations in the in-situ CH₄ enhancements in urban areas determined by ground-based high-precision measurements

For the $CH_4$ enhancements in urban and suburban areas, the urban BJ station exhibited the greatest

enhancements, with an annual mean enhancement ranging from 200–350 ppb over the 2019 season

(Fig. 5a and b), followed by the suburban XH station (with seasonal enhancements ranging from

50–200 ppb), compared with the concentration at the regional background SDZ station. The jumps

during the months shown in Fig. 5a occurred because the hourly means from 23:00–23:59 within a

certain month differed from those from 00:00–00:59 within the next month. The MLO data revealed

the lowest surface concentrations (~1800 ppb) and provided a global-scale background. The three

regional sites all exhibited obvious enhancements over the MLO. The $CH_4$ dome observed in northern

China is comparable to that observed in other cities in Canada and the USA, such as Los Angeles

(Verhulst et al., 2017) and Washington, D.C. (Huang et al., 2019), but higher than that observed in Salt

Lake City and Toronto, with values ranging from 100–1000 ppb (Mitchell et al., 2022). These surface

enhancements also exhibited seasonal cycles, with higher values in autumn (371 and 193 ppb at BJ and

330 XH, respectively) and lower signals in spring (195 and 70 ppb at BJ and XH, respectively). Moreover,

the monthly enhancements were consistent with this trend, with high enhancements from September to

November (Fig. S3). We compared the enhancement at whole time series (00:00-23:59) with the

afternoon well-mixed period (14:00-16:00), and the annual mean differences between them are 59.1

ppb for Beijing and 62.5 ppb for XH (Fig.S4), respectively.

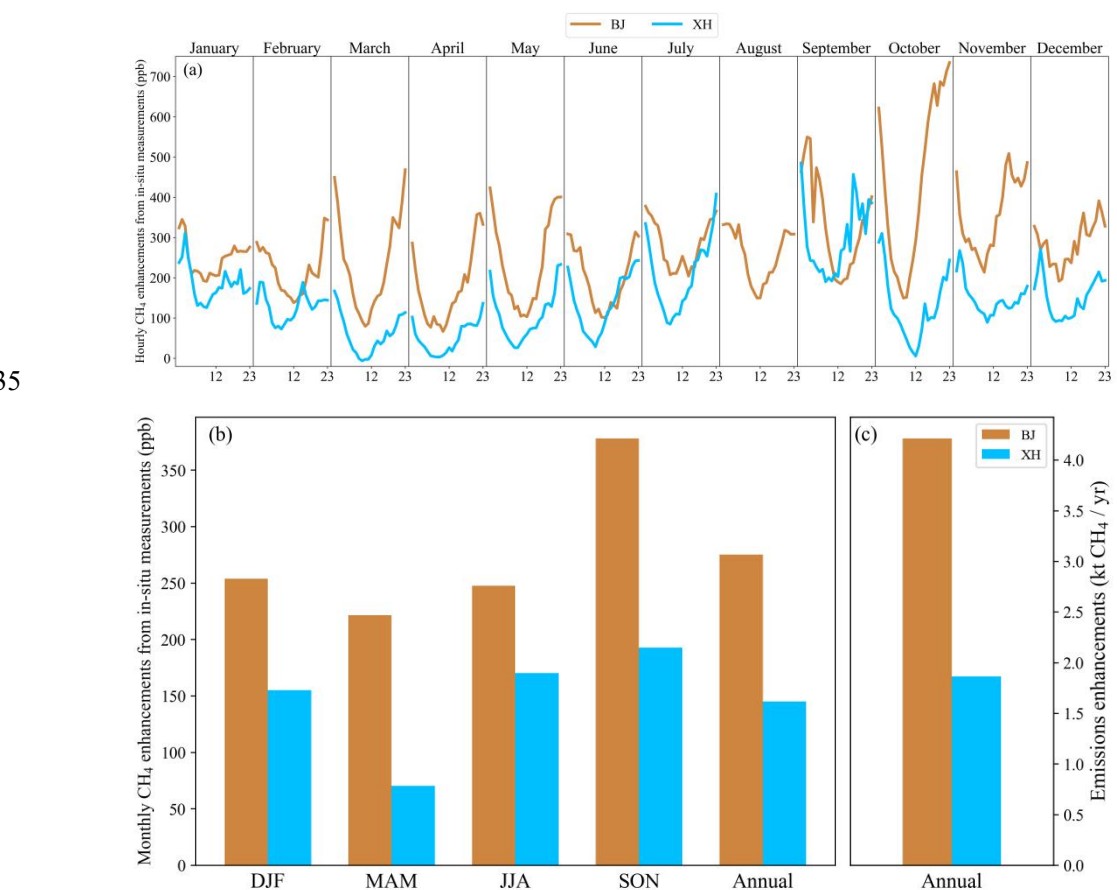

**Fig. 5 Enhancements in the in situ $CH_4$ concentrations and emissions at BJ and XH compared with that at SDZ for the hourly mean from 00:00–23:59 (UTC time) in each month (Panel a) and during the four seasons and for the annual mean (Panel b) in 2019.**

Furthermore, to explore the high-resolution and variations in time (Chen and Prinn, 2005; Rivera Martinez et al., 2023), we showed the enhancements in the in-situ CH₄ concentrations at BJ and XH compared with that at SDZ. For the daily-monthly (dots-line) mean enhancements from 2019-2020 (Fig. 6a), the daily enhancements ranged from 0-1200 ppb and the monthly enhancements ranged from 0-600 ppb during 2019-2020, and BJ enhancements were much higher than XH. Both sites showed

higher enhancements in autumn and winter days. For the hourly to daily enhancements from April 1st to June 30th 2019 (Fig. 6b), the hourly enhancements reached 2000 ppb at polluted events, and daily enhancements ranged 0-500 ppb.

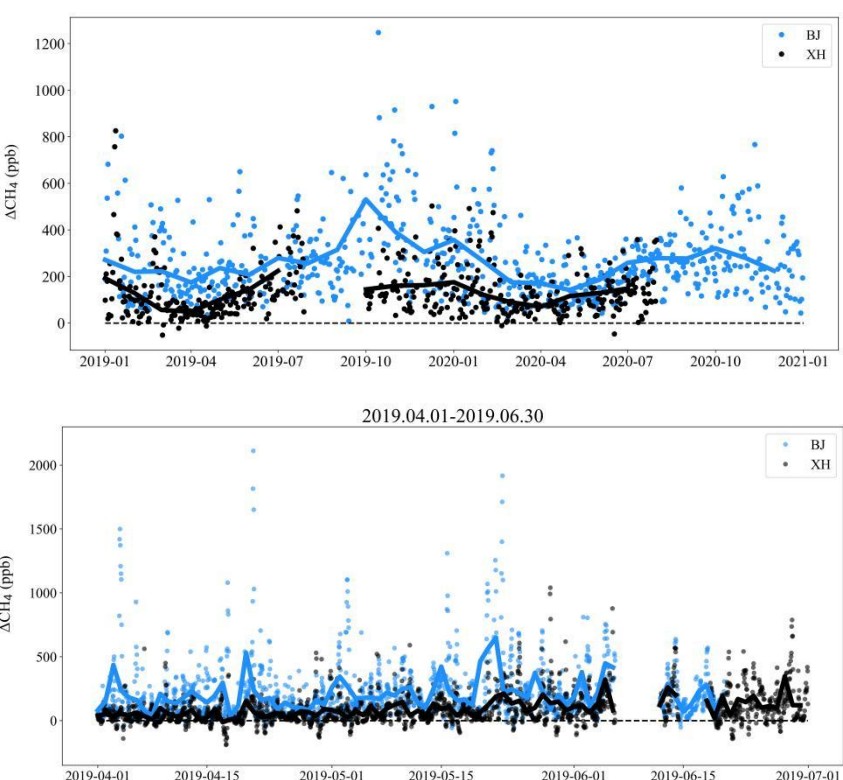

**Fig. 6 Enhancements in the in situ CH₄ concentrations at BJ and XH compared with that at SDZ for the daily-monthly (dots-line) mean (a) from 2019-2020, and hourly-daily (dots-line) mean (b) from April 1st to June 30th 2019 to show the high-resolution and variations in time. Data gaps were due to instrument malfunctions.**

**3.2 Correlations between the satellite-based XCH₄ concentrations and surface observations**

Satellite observations have an advantage in spatial coverage, yet they need careful calibrations and validations, especially for regional scale studies. Surface concentrations are more influenced by ground emissions, yet generally have good relationships with column concentrations (Fig. S22) (Ialongo et al., 2020). The satellite and surface observations generally agreed well in capturing seasonal variations, the

CH₄ concentration was highest in autumn, and the concentration decreased to a low level in winter (Fig.

7). The phase of the cycles in the seasonal column CH₄ concentrations at BJ, XH, and SDZ from the

TROPOMI data was consistent with that of the surface in situ measurements at the monthly scale (Fig.

7a, b and Fig. S3). However, the in situ CH₄ concentrations were greatly influenced by local emissions

and meteorological conditions, with larger amplitudes than those of the TROPOMI data, thus yielding

lower correlations with XCH₄ ($R^2$=0.16 and 0.48, for $p$<0.05 and 0.01 at BJ and SDZ, respectively; Fig.

7a, b). Furthermore, the urban BJ station exhibited higher XCH₄ values than those at the suburban XH

station and the regional background SDZ station, which is consistent with the surface in situ

measurements with higher signals. These results also indicated high local anthropogenic emissions in

the urban areas of Beijing. At HF, the average XCH₄ was highest from August–September because of

emissions originating from surrounding wetlands and rice paddies (Figs. 4d and 1). BJ and SDZ

exhibited the highest values from September–October.

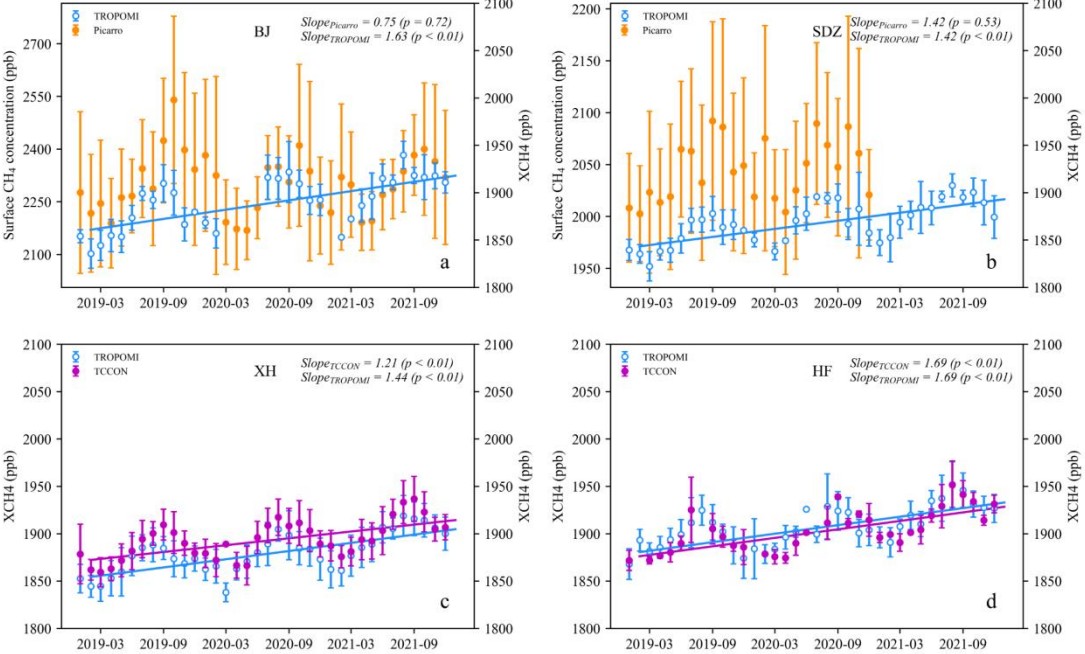

**Fig. 7 Temporal variations in the mean monthly XCH₄ and surface CH₄ concentrations observed at the BJ (a), SDZ (b), XH (c), and HF (d) stations from 2019–2021. Note that the scales differ between BJ and SDZ. And the two y-axis plots in (a-b) are with different scales (higher for Picarro). BJ, XH, SDZ, and HF**

**represent Beijing, Xianghe, Shangdianzi, and Hefei, respectively.**

As expected, the phase and magnitude of the seasonal cycles observed in the TROPOMI data at XH

and HF better agreed with those in the TCCON data than did the in situ comparisons (Fig. 8), with

higher variations in the in situ observations (Fig. 8a, b, S11). The $R^2$ values were 0.76 and 0.75 for XH and HF (Fig. 8c, d), respectively, which are higher than those with the in situ observations (Fig. 8a, b),

with a $p$ value <0.01 at both sites. Moreover, a daily scale comparison showed much weaker correlations (Fig. S11-S13). Furthermore, the surface $CH_4$ concentrations at XH site observed by Picarro showed differences when retrievals were successful and not for TCOON (Fig. S14). And when retrievals were not successful, the surface $CH_4$ concentrations usually reached peaks. The reduced air pressure was not considered in the comparisons across different sites (Beijing, Xianghe, Shangdianzi,

and Hefei), and the elevations of these four sites were less than 500m, which is within the boundary layer and mixed well for $CH_4$ concentrations. We also acknowledge that it is better to consider this factor when comparing with the higher elevation observations such as in Shanxi sites.

Moreover, there were seasonal trends between the TROPOMI and TCCON observations, with positive biases in the TROPOMI observations in spring and summer (5–15 ppb, or 0.5%) and negative biases in

winter (~-5 ppb, or 0.25%) compared with the TCCON observations, which is consistent with the findings of Sha et al. (2021). We further analyzed the bias at three resolutions at 0.1, 0.25, and 2.5 degree (Fig. S10), and the mean bias increased with the TROPOMI resolution, from -4.2 to -13.8 ppb in XH, and -0.7 to 4.4 ppb in HF, but lower resolution would match less TROPOMI observations, indicating a moderate resolution is needed. The TROPOMI $XCH_4$ and Hefei TCCON

$XH_2O$ values were significantly positively correlated, with $R^2 = 0.43$ and a $p$ value <0.01 (Fig. 9b). Similar results have also been reported in other studies. High $CH_4$ biases at high latitudes correlated with $H_2O$ columns were found in $H_2O$ retrievals from the TROPOMI by Schneider et al. (2020) and Lorente et al. (2021). The satellite retrieval biases might also be associated with cloudiness in summer and thus the limited number of TROPOMI observations (Qu et al., 2021) and relevant surface albedo

and scattering issues (Barré et al., 2021; Schneising et al., 2023; Schneising et al., 2019). Moreover, we calculated the $XCH_4$ and in situ $CH_4$ growth rates. The $XCH_4$ level observed from the TROPOMI data clearly increased from 2019–2021, with increase rates ranging from 1.4 to 1.6 ppb month$^{-1}$ ($p < 0.01$, Fig. 7). Future TROPOMI validations for potential $H_2O$ impacts need vertical profile observations in southern area (e.g. in Anhui and Henan Province) in summer.

Furthermore, the satellite platforms measure radiance, which is then used to invert or constrain a column concentration loading. There are additional assumptions and steps required to go from

concentration to emissions. And both steps introduces uncertainties and errors. Several recently published paper demonstrated that the later step can lead to extremely different emissions end point inversions, especially at high spatial and temporal resolution (Guanter et al., 2021; Pei et al., 2023; Qin et al., 2023). In order to assess the precision and uncertainty, several errors are needed to be considered: measurement errors, parameter errors, approximation errors, resolution errors, and system errors (Povey and Grainger, 2015). And the use of ensemble techniques (e.g. different algorithms or forward models) to present multiple self-consistent realizations of a data set is useful in depicting unquantified uncertainties. Furthermore, although the inversion algorithms and related processing methods of TCCON and TROPOMI are different, TCCON, as a calibration standard for docking with WMO, can be used as satellite authenticity test data, and the comparison between TCCON and TROPOMI can be used to transmit WMO standards to satellite observations.

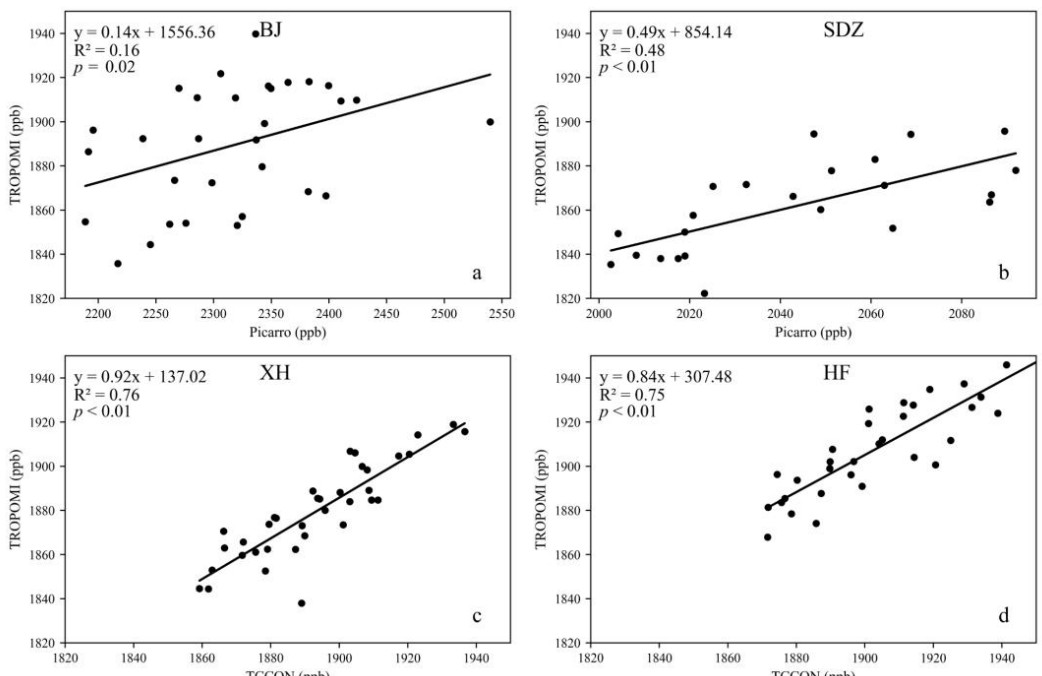

Fig. 8 Correlations between the mean monthly $XCH_4$ concentration from the TROPOMI dataset and the surface $CH_4$ concentration observed at BJ (a) and SDZ (b) and correlations between the mean monthly $XCH_4$ concentrations from the TROPOMI and TCCON datasets at XH (c) and HF (d) from 2019–2021. BJ, XH, SDZ, and HF represent Beijing, Xianghe, Shangdianzi, and Hefei, respectively.

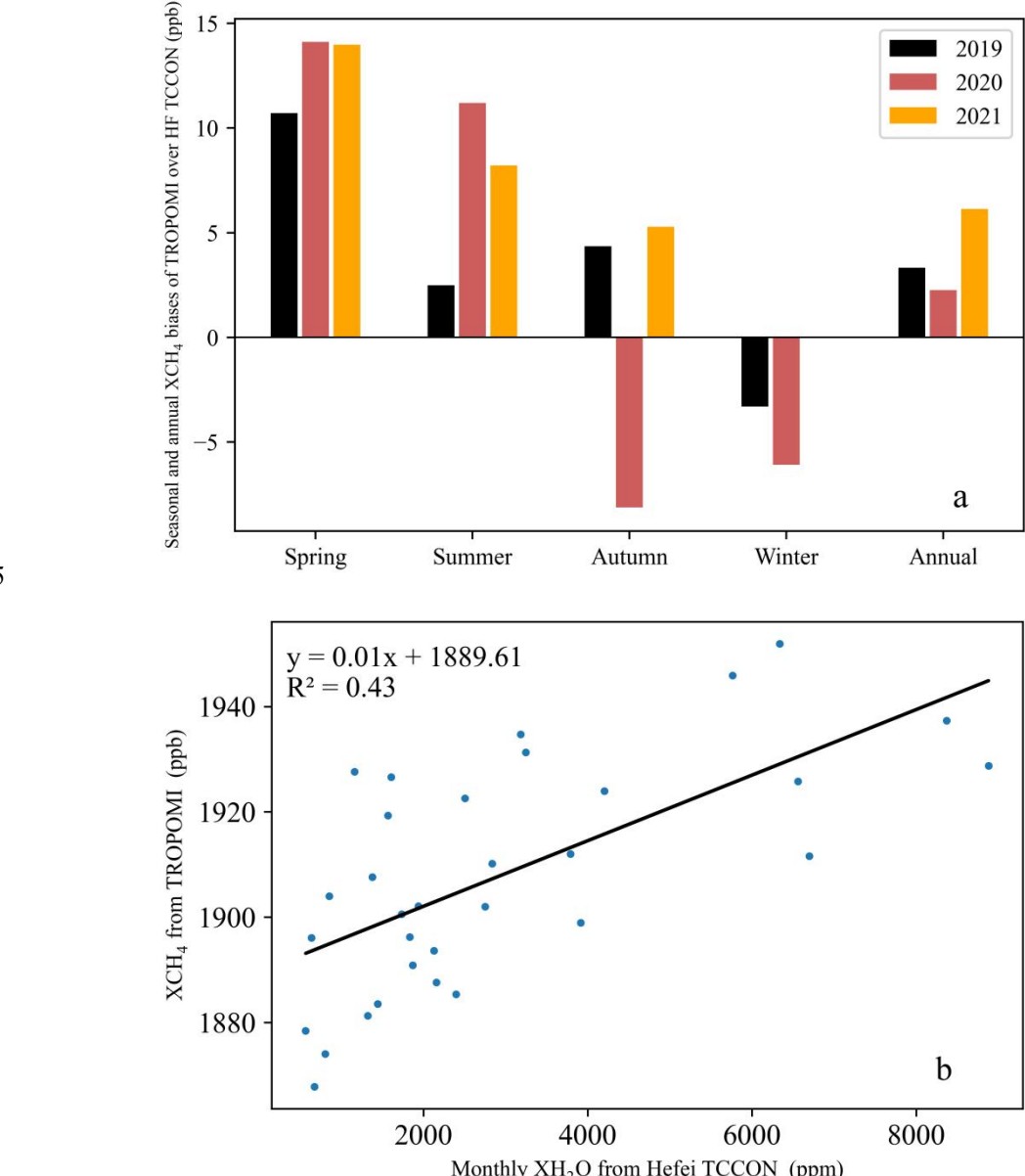

**Fig. 9 (a) Seasonal and annual biases of the TROPOMI XCH₄ observations minus the Hefei TCCON observations from 2019–2021. (b) Positive correlations between the monthly XH₂O from the Hefei TCCON observations and XCH₄ from the TROPOMI observations.**

**3.3 Validation of the model performance against in situ measurements**

To assess the accuracy of the posterior CH₄ data from the GEOS-Chem and CAMS models, we compared the simulated concentrations with the high-precision measurements at BJ, XH, and SDZ. We used the mean bias (MB) and root mean square error (RMSE) to assess the model performance. The MB is an indicator of systematic biases, while the RMSE reflects the spread in simulations with higher weights for large errors. In general, the models captured the CH₄ trends and variations (Figs. 7 and 8, respectively). The daily comparisons between the simulations and observations revealed a negative bias

at the urban BJ site (MB=-57.2 ppb) and positive biases at the XH (77.4 ppb) and SDZ (68.0 ppb) sites (Fig. 10). The optimized GEOS-Chem model captured the observed baseline at BJ (Fig. 10a) but slightly overestimated the baselines at XH and SDZ (Fig. 10b, c and Fig. 11). Moreover, the RMSEs

for the three sites decreased from the urban BJ site (185.6 ppb) to the suburban XH site (157.7 ppb) and the regional background SDZ site (110.7 ppb) (Fig. 11). The simulations could not capture some of the peak values in urban Beijing (Fig. 10a), indicating considerable simulation challenges in urban areas with complex anthropogenic emissions, which is consistent with the $CO_2$ and air pollutant simulations (Feng et al., 2019; Liang et al., 2022). And this requires a high-resolution $CH_4$ simulation and inversion

system to assimilate the observed urban-rural gradient.

The seasonal and annual biases of the GEOS-Chem and CAMS models relative to the in situ measurements were calculated (Fig. 11, S16). Both models showed negative biases (-39 ~ -152 ppb) from spring to autumn in Beijing and positive biases (17~86 ppb) in winter, which was due to the higher baseline simulations on clean days (Fig. 11a). The GEOS-Chem model simulations revealed

positive biases in the annual mean at the suburban and regional background sites during most seasons (Fig. 11b, c). The CAMS model showed positive biases at XH and negative biases at SDZ, and both sites showed positive biases in winter (Fig. 11b, c). The hourly and monthly comparisons also revealed similar variations and trends (Figs. S5 and S6, respectively), while the monthly and seasonal data from the CAMS model also revealed negative biases in urban areas and positive biases in suburban areas,

with lower biases at the suburban and regional background sites than at the urban site (Figs. 8c and S6). As researched by (Zhang et al., 2024) and in this study, the outdated a priori emissions datasets indeed introduced errors in both spatial distribution and magnitude, and the temporal variation, which introduced errors in forward transport simulation and thus inversions (Fig. 10-11, biases at three sites). These errors could be largely adjusted by the data assimilation algorithm, but more accurate a priori

could induce less errors in the data assimilation and produce more accurate posterior estimates. (Yu et al., 2021) described the impact of errors in prior estimates on inversions results. And they showed that 4D-Var analysis of the TROPOMI data improved monthly emission estimates at 25 km even with a spatially biased prior.

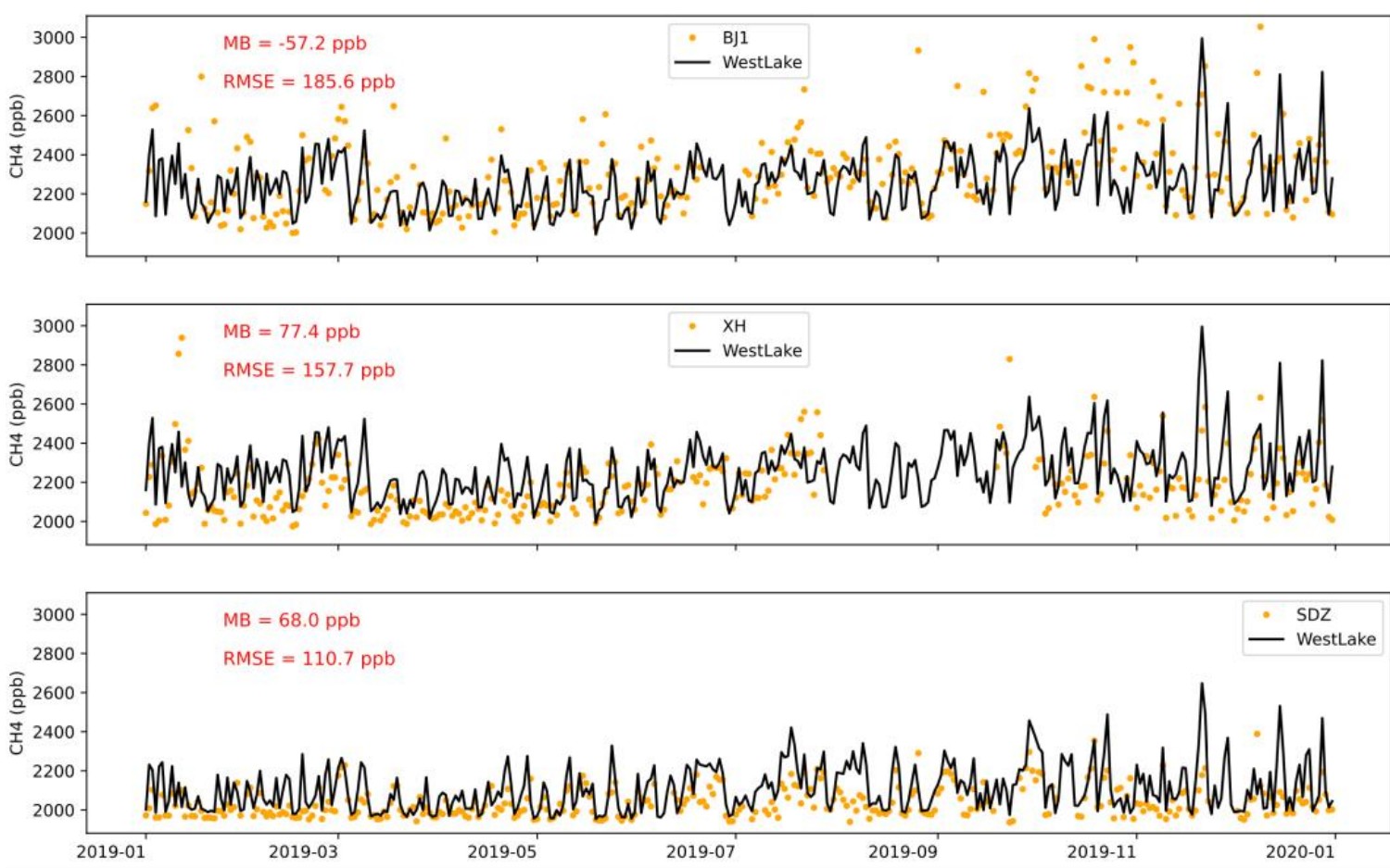

Fig. 10 Daily comparisons of the Westlake model simulations with the in situ high-precision measurements at the three sites. MB denotes the mean bias, and RMSE is the mean root square error. BJ, XH, and SDZ denote the observations from the Beijing, Xianghe, and Shangdianzi stations, respectively.

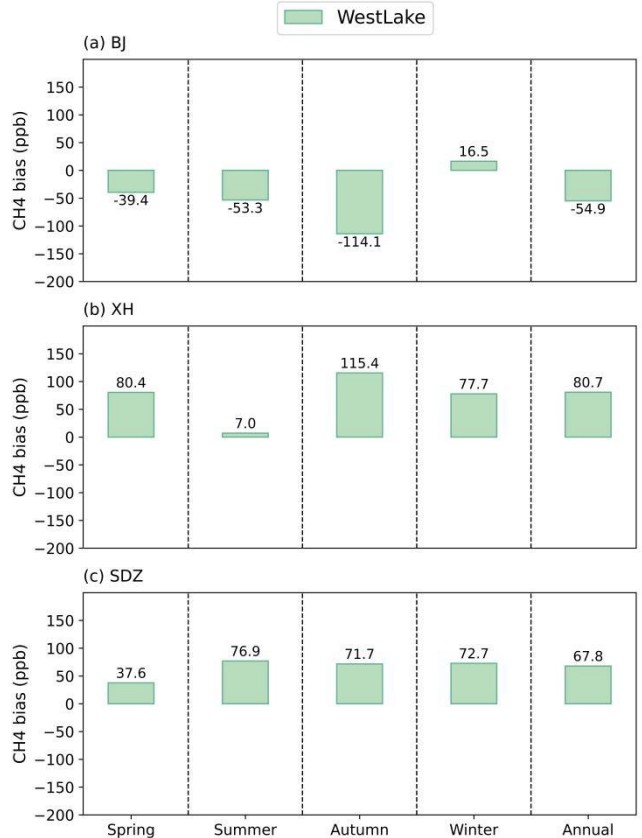

**Fig. 11 Seasonal and annual biases for the Westlake model compared with the high-precision in situ measurements at BJ, XH, and SDZ. BJ, XH, and SDZ represent Beijing, Xianghe, and Shangdianzi, respectively.**

### 3.4 Spatial characteristics of the inversion-optimized surface CH$_4$ concentrations and their correlations with emissions

Northern China serves as a notable CH$_4$ source, and the total emissions increased to 14.2 Tg yr$^{-1}$ in 2019 (Figs. 1, 15 and S2), accounting for 30% of the total emissions in China (Fig. S25) (Liu et al., 2021b). Correspondingly, the optimized GEOS-Chem model surface CH$_4$ concentration reached 2112.1 ppb in 2019. High surface concentrations were mostly consistent with high emissions in southern Shanxi (mainly coal mine emissions), northern Henan, central-northern Anhui, southern Hebei and Beijing (Figs. 12–13, S17-S18, and Fig. 1), which has also been reported by Peng et al. (2023), Qin et al. (2023), and Han et al. (2024) in Shanxi. These enhancements are not only located in large urban areas, but also located in coal and oil productions (Fig. 12-13 and Fig. S17-18). In contrast, most

parts of Shandong, Jiangsu and northern Hebei indicated a relatively low annual mean concentration in 2019. In terms of seasonal and monthly spatial patterns, spring (March–April–May) exhibited the lowest values, while autumn (September–October–November) and winter (January–February) exhibited much higher concentrations (Figs. 12 and 13, respectively). The northern part demonstrated lower concentrations in winter than in summer, largely due to the favorable climatic conditions for dispersion, with greater wind speeds in winter than during the other seasons (Figs. 9 and 10). We also compared the Westlake column $CH_4$ data with the TROPOMI observations, and these two datasets exhibited consistent spatial patterns across all seasons (Fig. S19) and suitable correlations ($R^2$=0.46~0.74, $p < 0.01$; Fig. S20).

Moreover, the spatial patterns of the surface and column concentrations revealed consistent trends in spring (March–April–May) and summer (June–July–August) (Figs. 12-13 and S19-S22, respectively), and the $R^2$ values for the correlations during four seasons were 0.46-0.74 for TROPOMI $XCH_4$ and Westlake $XCH_4$ (Fig. S20). Moreover, we selected three high concentration (emitting) days in 2019 (Fig. S23). Although there were systematic errors between TROPOMI and Westlake modeled results, they showed generally consistent spatial patterns, with much higher concentrations in southern Shanxi and Henan, and lower concentration in northern Shanxi and northern Hebei. And the R even reached 0.6-0.7 for two cases, which is consistent with studies in TROPOMI $NO_2$ (Ialongo et al., 2020; Verhoelst et al., 2021; Wang et al., 2020).

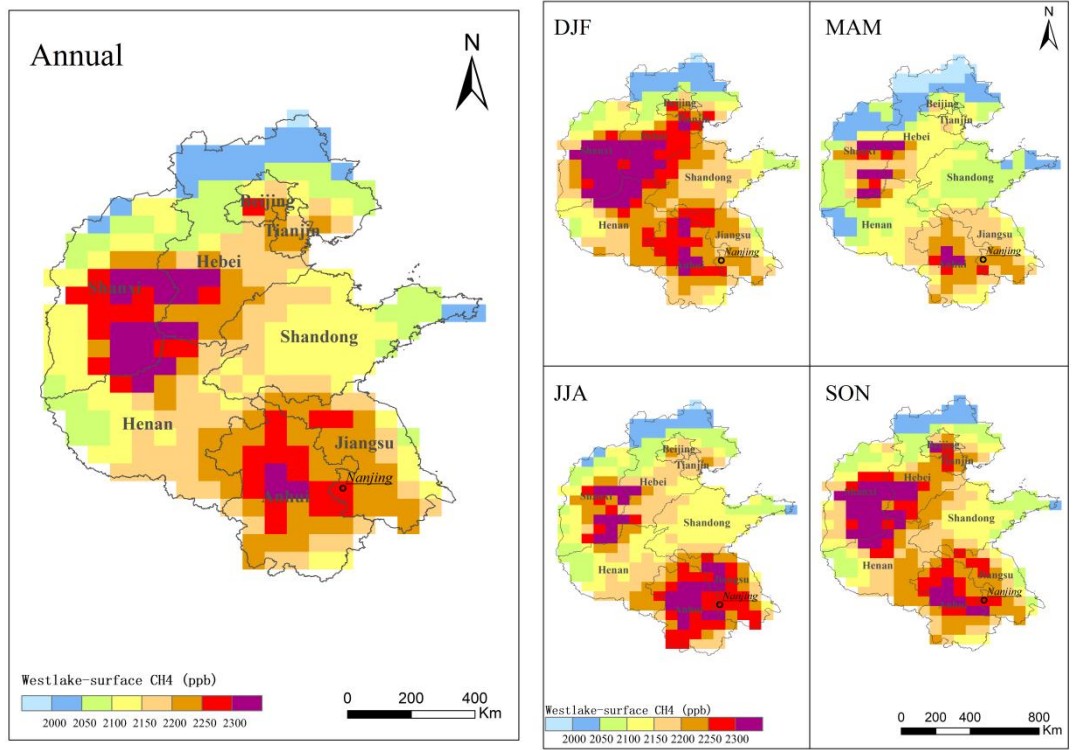

**Fig. 12 Posterior annual and seasonal spatial distributions of the surface CH₄ concentration (Westlake model) in northern China in 2019. For the Annual panel, purple colors were used to show the coal mining emissions, which is generally consistent with the high concentration regions.**

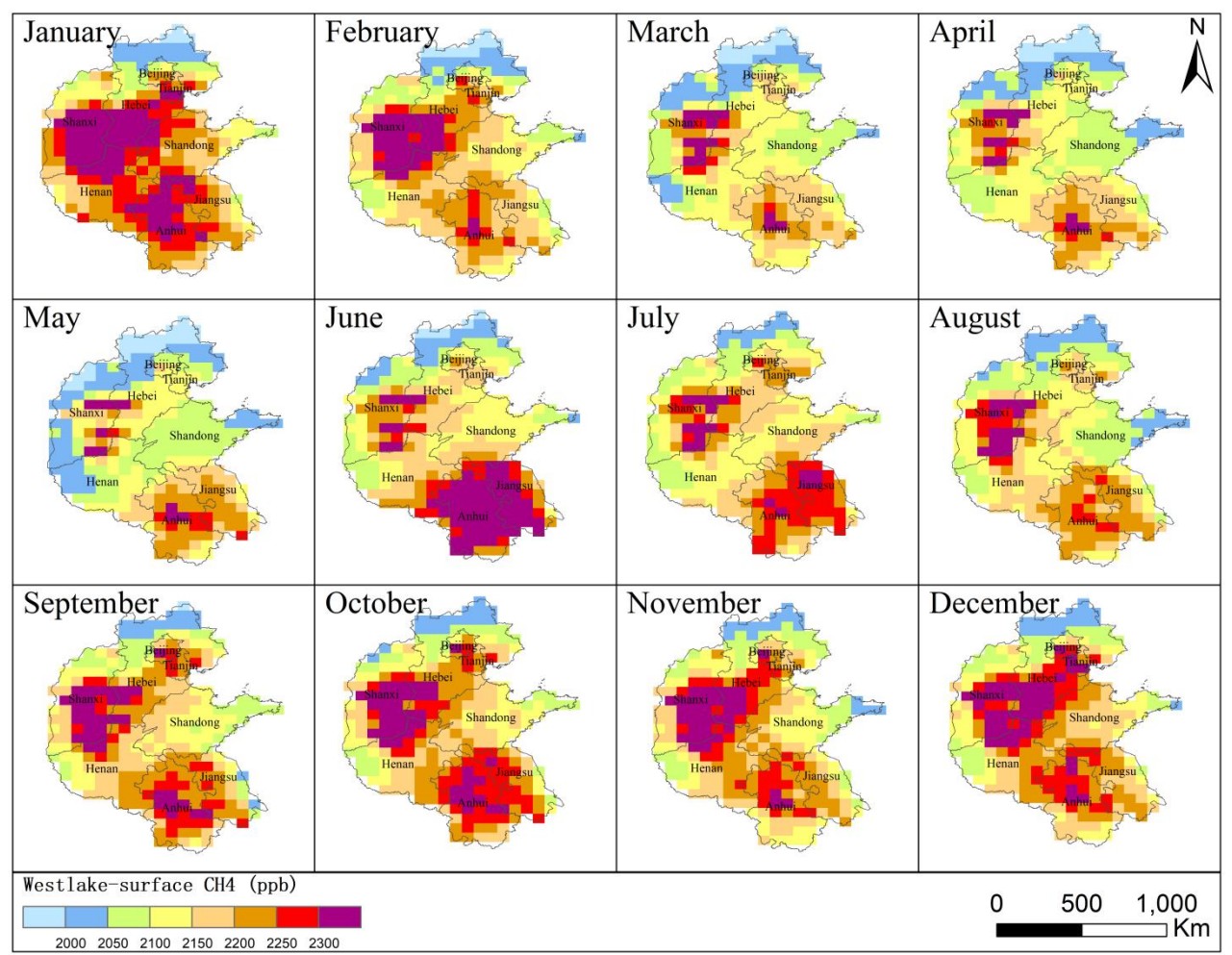

**Fig. 13 Posterior monthly spatial distributions of the surface CH₄ concentration (Westlake model) in northern China in 2019.**

Enhanced surface $CH_4$ concentrations are an indication of high local emissions associated with coal mining, oil and gas activities, agriculture, or wetland processes (Barré et al., 2021; Gouw et al., 2020; Zhang et al., 2020). The correlation between the PKU-v2 emissions and the simulated surface concentrations at the grid level (Fig. 14) was statistically significant ($p<0.05$), albeit with a low $R^2$ value of 0.20. Regressions with or without 2 outliers or different emissions intervals also showed similar positive correlations (Fig. S24). Positive relationships with low $R^2$ have also been found in oil and gas production regions in the U.S. (Gouw et al., 2020).

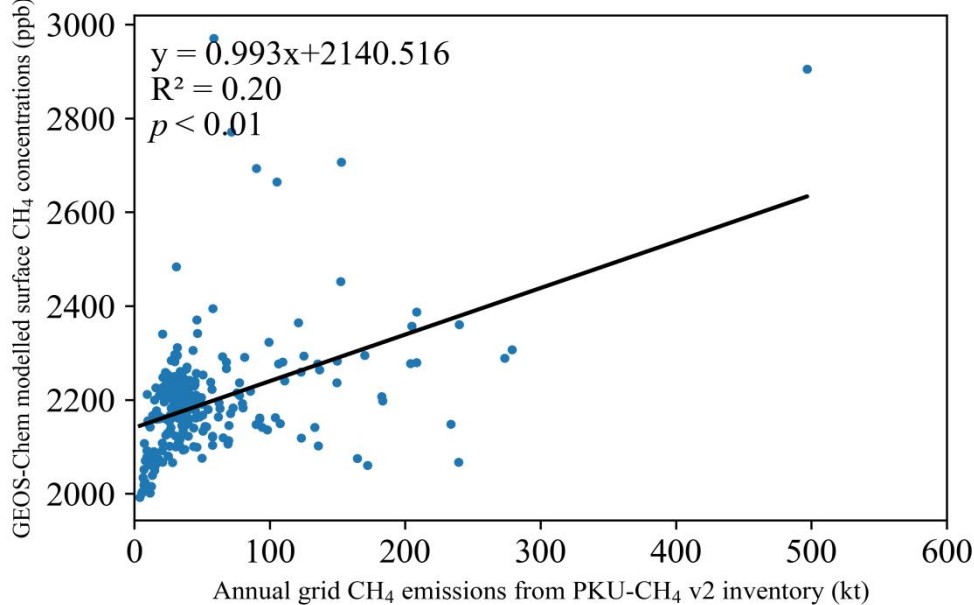

**Fig. 14 Correlations between the $CH_4$ emissions from the PKU-$CH_4$-v2- and Westlake modeled surface $CH_4$ concentrations. PKU-$CH_4$ data (10 km resolution) were aggregated to Westlake-model resolution (50 km) to keep consistency.**

Moreover, we found a positive correlation between the Westlake column $CH_4$ and PKU-$CH_4$-v2 emission inventories (Fig. S24), which demonstrates the ability of satellite data to detect large local sources, such as coal mining (Peng et al., 2023), in urban regions. Recent studies have also indicated that TROPOMI data combined with other satellite observations can be employed to identify large emission sources in gas and oil well blowouts (Cusworth et al., 2021; Schuit et al., 2023). These findings showed the potential use of satellite data in detecting hotspot emissions that may be omitted from emissions inventory, and thus further mobile observations (e.g. cars and UAVs) can be used for field double check in environmental enforcement.

**3.5 CH4 emission estimates from inventories and inversions**

The CH4 emissions were 14.3, 28.5, and 24.0 Tg for PKU-v2, EDGAR-v4.3.2, and Westlake model,
respectively (Fig. 15a). The posterior results from Westlake model using TROPOMI as observations
showed a 15.6% (4.4 Tg) decrease than the prior (EDGAR v4.3.2). Emissions were adjusted for higher
values in Hebei, Jiangsu, and Anhui (Fig. 15d, h, i), and were adjusted for smaller values in Shanxi,
Shandong, and Henan (Fig. 15e, f, g), which were consistent with spatial pattern of TROPOMI XCH4
(Fig. S19). A literature review by this study showed a range of 4.4-13.1 Tg CH4/yr for Shanxi CMM
emissions (Table S2) (Chen et al., 2024; Janssens-Maenhout et al., 2019b; Kang et al., 2024; Peng et al.,
2023; Qin et al., 2024; Sheng et al., 2019a; Tate, 2022). And the rest emissions from this region showed
a range of 6.6-20.1 Tg, resulting in a range of 11.0-28.5 Tg for total emissions (Table S2)
(Janssens-Maenhout et al., 2019b; Peng et al., 2016). We thus recommend more researchers make their
datasets publicly available, more in-situ observations progressed in this region, and high
spatial-resolution modelling studies conducted at city to province level with multiple sources
observations.

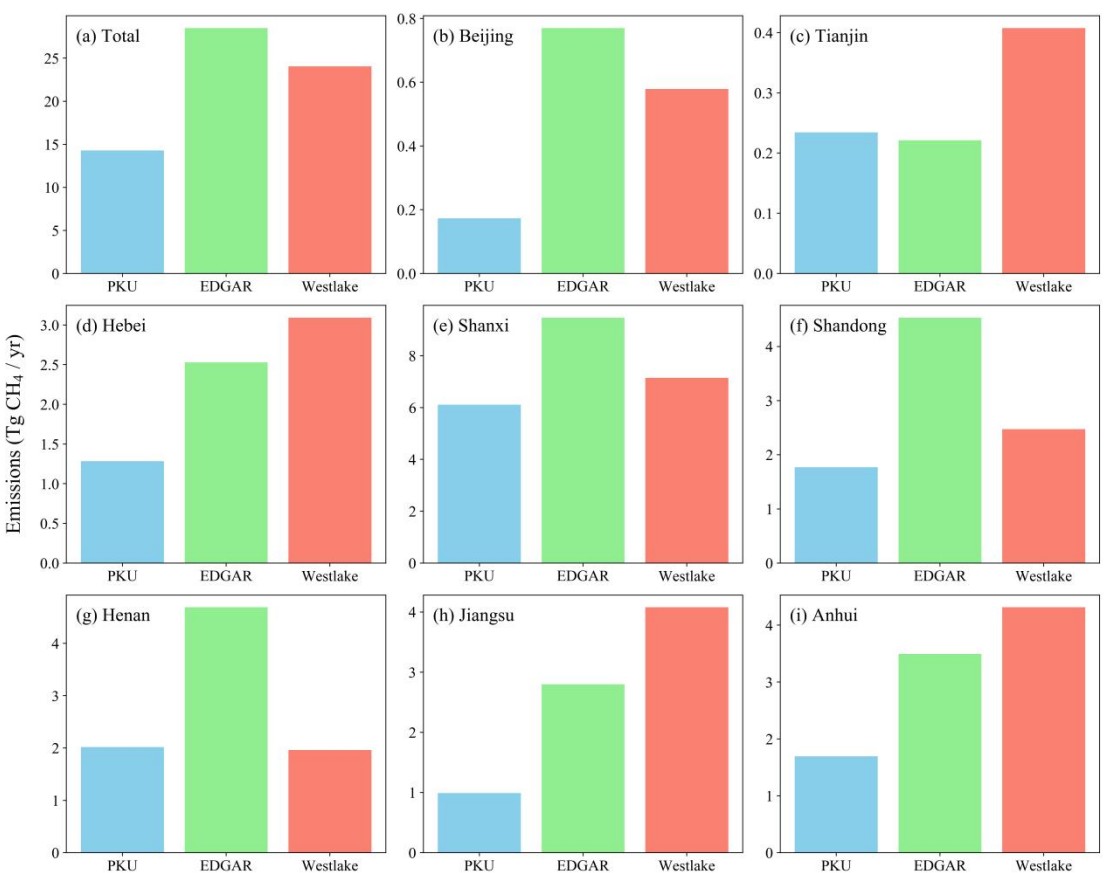

**Fig. 15 Total and provincial CH$_4$ emissions from PKU-v2, EDGAR-V4.3.2 inventories and Westlake model**

**posterior results. Note that the scales are different for provinces.**

## 4 Conclusions

In this study, we compiled a comprehensive dataset to study the spatiotemporal characteristics of surface and column CH$_4$ concentration variations and their correlations with emissions. We found that surface CH$_4$ concentrations can be much greater (500–1500 ppb) than regional background CH4

concentrations in urban and suburban areas because of anthropogenic emissions. Notable seasonal surface enhancements of 200–350 ppb at urban Beijing station and 50–200 ppb at the suburban Xianghe station were observed compared with the concentration at the regional background Shangdianzi station. Positive relationships were found between the surface (both in situ and TCCON) and TROPOMI column observations. The inversion-optimized concentrations generally agreed well

with the surface in situ observations in terms of the seasons and annual means in 2019. A generally spatial-consistent pattern was observed between the posterior results and the Tropospheric Monitoring Instrument (TROPOMI) column CH$_4$ observations for all seasons and annual mean. The optimized surface CH$_4$ concentrations were relatively high in southern Shanxi, northern Henan, and Beijing (with enhancements of ~300 ppb), whereas relatively low concentrations were observed in northern Hebei

and most parts of Shandong, which was positively correlated with the PKU-CH$_4$-v2 emission inventory data. The posterior results using TROPOMI observations was 24.0 Tg CH$_4$, a decrease of 15.6% (4.4 Tg), compared with the prior EDGARv4.3.2. This study provides a comprehensive dataset of CH$_4$ concentrations and spatial gradients in northern China, which provides key data for further observations, high-resolution atmospheric inversions and policy-making related to emission reduction.

**Supporting information**

Sectoral CH$_4$ emissions at the provincial level and simulated posterior monthly mean CH$_4$ concentrations, enhancements, etc., are contained in the Supporting information.

**Data availability statement**

The data used to generate the figures in this manuscript are available as open-access data at https://doi.org/10.5281/zenodo.10957950 (Han et al., 2024).

**Competing interests**

The authors declare that they have no conflicts of interest.

**Funding**

This research was supported by the National Key R&D Program of China (No. 2023YFC3705500 and 2017YFB0504000), the China Quality Certification Center project—Monitoring, simulation and inventory joint assessment of carbon emissions in typical industrial parks against the background of carbon peaking and carbon neutrality (grant no. 2022ZJYF001), the Qiluzhongke Institute of Carbon Neutrality Program of Jinan Dual Carbon Simulator, and the Chinese Academy of Sciences (CAS)

Proof of Concept Program—Carbon neutrality-oriented urban carbon monitoring system and its industrialization (grant no. CAS-GNYZ-2022).

**Author contributions**

PFH, QXC and WQS conceived and designed the study. PFH, QXC, RSL, WQS, and SXL collected and analyzed the datasets. PFH led the writing of the paper, with contributions from all coauthors. All

585 coauthors contributed to the descriptions and discussions in the manuscript.

**Acknowledgments**

We thank all the team members involved in the surface in situ observations. We thank Professor Shushi Peng for kindly providing the emission inventory and suggestions for the manuscript. We thank Dr. Xiaohui Lin and Mr. Zhoutong Liang for their help in the data analyses.

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
