# Peer review of "Spatiotemporal variations in atmospheric CH4 concentrations and enhancements in northern China based on a comprehensive dataset: Ground-based observations, TROPOMI data, inventory data and inversions"

_EGUsphere, 2024_

## Author Response (AR1)

**Reviewer #1**

This study investigates methane (CH4) emissions in Northern China, a region recognized for its significant contribution to global greenhouse gas emissions. Although CH4 is a potent greenhouse gas, uncertainties surrounding CH4 emissions persist, particularly in urban and industrial areas of Northern China. To address these uncertainties, the authors compiled a comprehensive dataset that integrates ground and satellite observations, inventory data, and modeling results. This dataset aims to analyze CH4 concentrations, enhancements, and their spatiotemporal variations in the region. High-precision in situ observations from locations such as Beijing and Xianghe revealed distinct cycles and variation, with concentration enhancements ranging from 500 to 1500 ppb. The findings indicate a significant upward trend in CH4 concentrations over time, corroborated by both ground and satellite observations. The study highlights that anthropogenic activities are major contributors to variations in surface CH4 concentrations, especially in middle and southern Shanxi Province and northern Hebei Province. However, discrepancies were noted between optimized simulations based on surface atmospheric inversion data and observations from the Tropospheric Monitoring Instrument (TROPOMI) during summer months, suggesting potential systematic biases. Their posterior analysis of CH4 concentrations revealed that areas such as southern Shanxi, northern Henan, and Beijing exhibited elevated levels (an increase of approximately 300 ppb), which correlated positively with existing emission inventory data. This research provides essential insights into CH4 emissions in high-emission areas of Northern China and offers a valuable dataset for researchers and policymakers aiming to enhance observational strategies and formulate effective emission reduction policies.

In my opinion this paper is a bit weak for ACP. However, since the paper is anchored in measurements, I would like to see it worked on and ultimately published in ACP. If the authors are willing to address the three general points listed here as well as specific details below, I believe that the standard can reach that of ACP and make a high quality and impactful paper.

Response: We thank you for your understanding of this paper. Indeed this paper focuses more on the observations. We made major revisions according to your suggestions. We added more analyses on in-situ observations for high-resolution in time (e.g. hourly to daily, Fig. 3-Fig.6). And we also did more spatial analyses on surface, column concentrations and emissions from TROPOMI, Westlake model, and Picarro observations (Fig. S8-S14, S19-20, S23). We added more comparisons with observations from Taiyuan, Shanxi, and USA cities at similar latitudes (Fig. 4). And

we analyzed the TROPOMI resolution effects on biases with TCCON (Fig. S10). During the deepen of these analyses, we found that we flipped the latitude in our previous figures which influenced the spatial maps (Fig. 9-11), and we revised all of these errors in this major revision version, which showed good consistency between emissions and surface concentrations (Fig. S17-18,S24).

(a)  The authors need to look at more literature and bring in more observations. Knowledge of where coal mine and oil well emissions are located on the ground does not currently match with the modeled emissions fields.

Response: We researched the literature again and listed the observations and emission estimates in Table S1 and S2. Indeed as you have pointed out, there is a large data gap in observations in this region, and the published paper are mostly not open-access for data. We corrected the code bug and revised Figs9-11, and the high concentrations are now very consistent with the $CH_4$ emissions. We further overlapped coal mine and oil well emissions with simulated surface $CH_4$. The high emissions generally matched with high emissions (Fig. 9-10, Fig.S9-10).

(b)  The authors demonstrate that there is extensive day-to-day variability in the observations. They should therefore find a way to evaluate the products more effectively. The way in which they compute month-by-month values and evaluate is out of date, as well as weakens the value of their high frequency observations.

Response: Thank you for this good suggestions. We further analyzed the data at monthly, daily, and hourly scale in both concentrations (boxplots of monthly data, and ) and enhancements (added in Fig. 3-6; Fig.S4-S5). Diurnal variations showed $CH_4$ concentrations reached peak at around 08:00 local time (Fig. 4, S7), due to the lower boundary layer and accumulation of emissions (Bao et al., 2020; Zeng et al., 2021).

(c) They need to do a bit more work on their underlying explanations of their findings. I found the data to be far more interesting than the conclusions and statements made. Response: Thank you. We discussed with several experts and added explanations for hourly-daily-monthly concentration and enhancement variations, and discussions in lines around 295, 315 and 353.

Detailed Questions/Points to consider:

1. What exactly does the author mean when they state urban emissions of CH4? Do they mean leaky pipes, non-fully combusted CH4 from cooking and water heating, CH4 from urban rubbish, or do they also mean indirect CH4 from the goods and products consumed in urban areas? Similarly, the CH4 from coal and oil are large, but seem to not be

included. What about large industrial sources which use coal, oil, or natural gas and continue to leak/emit CH4? I think that better and wider referencing and more extensive literature review can address this issue.

Response: Thank you for the careful review. After a thorough check, we found that we had flipped the latitude when we drew figures and we now fixed this bug and re-drew all spatial-related figures (Fig 9-11). For the emissions, we mean the direct emissions from emission sources but not indirect emissions (added descriptions in lines around 80), e.g. including leakage/non-fully combustion from energy storage, transportation and consumption, landfills and waste water. These urban direct sources contribute to urban enhancements over the rural background (e.g. Fig.9 Beijing and Nanjing). Indeed, $CH_4$ from coal and oil production are large, but are generally not from urban areas (e.g. in coal mine areas in Shanxi province), and thus also have obvious enhancements in coal mine areas (Fig. 9). We added the corresponding analyses (Fig. 12-13, Fig.S17-18) and references in lines around 505.

2.  What is the impact from outdated a priori emissions datasets? Is it related to the magnitude being wrong, the spatial distribution being wrong, the temporal variation being wrong, or some combination of these factors together? (Zhang et al., 2024)

Response: Thank you for this question, and we added discussions in lines around 480. As researched by (Zhang et al., 2024) and in this study, the outdated a priori emissions datasets indeed introduced wrong message in both spatial distribution and magnitude, and the temporal variation, which introduced errors in forward transport simulation and thus inversions (Fig. 8, biases at three sites). These errors could be largely adjusted by the data assimilation algorithm, but more accurate a priori could induce less errors in the data assimilation and produce more accurate posterior estimates. (Yu et al., 2021) described the impact of errors in prior estimates on inversions results. They showed that 4D-Var analysis of the TROPOMI data improved monthly emission estimates at 25 km even with a spatially biased prior.

3.  These satellite platforms measure radiance, which is then used to invert or constrain a column concentration loading. This step involves many assumptions and approximations, and produces uncertainty. There are additional assumptions and steps required to go from concentration to emissions. This second set of steps introduces further uncertainties and errors. In fact, there is a lot of literature recently published which demonstrates that this later step is not error free or fast and simple, and can lead to extremely different emissions end point inversions, especially at high spatial and temporal resolution. (Guanter et al., 2021; Pei et al., 2023; Qin et al., 2023).

Response: Thank you for this good point. We added these uncertainty discussions and literature around lines 430.

4.  Are you sure that only 30% of anthropogenic CH4 from China was from your region of interest? I would think that the coal based CH4 emissions from Shanxi and Anhui and Oil based CH4 emissions from Shandong would be close to this number on their own, based on some newer studies. Only 5.7Tg/yr of emissions from Shanxi seems too low. Perhaps

characterizing a range of values across the literature would be more effective rather than relying on percentages? (Qin et al., 2024).

Response: Thank you for this question and we added more analyses (lines around 570) and literature review (Table S2, Fig. 14, Fig. S25). This 30% data is for the year 2010 emissions from PKU-CH$_4$-v2. And we revised according to your suggestions to include more study results. We plotted both the sectoral and total emissions and fractions of this region to the whole mainland China at Fig. S25. The literature review showed a range of 4.4-13.1 Tg CH$_4$/yr for Shanxi CMM emissions. And the rest emissions in this region were estimated as 6.6-20.1 Tg from PKU-v2, resulting a total range of 11.0-28.5 Tg emissions.

5. What is meant by a methane enhancement? What is the background value used? Does this include instrument minimum reliable observations including uncertainty, or just some other statistical technique? Since concentrations and total column loading are influenced by height, how is this corrected for in terms of deciding where is background? On top of this, how is background considered when making monthly or seasonal plots and comparisons? Day-to-day would be easier to achieve. (Chen and Prinn, 2005; Rivera Martinez et al., 2023).

Response: The methane enhancement is determined by the city urban site (Beijing) and suburban site (Xianghe), and the background is the WMO/GAW regional background Shangdianzi. Within the boundary layer, we considered the concentrations are mixed well, especially at 14:00-16:00 local time. We compared the enhancement at whole time series with the afternoon well-mixed period (14:00-16:00), and the annual mean difference is 59.1 ppb for Beijing and 62.5 ppb for XH, respectively (Fig.5 and S4, lines around 352). We further plotted day-to-day results in Fig.6. For most of time, SDZ showed a good background. And indeed, the enhancement calculated for SDZ as background showed a very small fraction of negative values, indicating a not real background. And we discussed this around lines 370.

6. All three of your surface stations are near to sea level. Are these representatives of the coal emitting regions in Shanxi, which are found from 1000m to 2000m above sea level. What approach is used to **account for this elevation difference**? (Li et al., 2023)

Response: Thank you for pointing this. In this study, we do not have in-situ observations in Shanxi, but we compared our results with the observations in Xiaodian, near Taiyuan, Shanxi. Through comparison with (Hu et al., 2023), and the hourly concentration ranged from 2000-6000ppb, with enhancements 0-1800ppb, which is comparable to the results in this study (Fig.2-3, and Fig.6). Moreover, the elevation was considered in the GEOS-Chem model with 47 layers (Liang et al., 2023).

7. All three of your surface stations also observe ranges of CH4 concentration which is far lower than high values observed in and around coal and oil areas in Shanxi, Anhui, and Shandong. Will this make a difference when doing inverse modeling? (Liu et al., 2022)

Response: Indeed, this will cause a difference when doing inverse modeling, but most observations are not publicly available, which hampers the inverse modeling, and that's why we made our observations publicly available. However, since this paper focuses more on the concentration datasets we generated and shared, further more inversions with high spatial resolution are still in progress and beyond the scope of this study. We added discussions in lines around 572.

8. Given the very strong North to South gradient in CH4 background concentration in the Northern Hemisphere (as observed at GAW stations and from AGAGE), would it make sense to use the data from Mona Loa at 19oN to compare to the background at stations located around 40oN? (Prinn et al., 2018)

Response: Thank you for this suggestion. The regional background is more relative to the local signals, and we used the regional background SDZ for the enhancement analyses, not the global background. Following your suggestion, we compared with observations in Shanxi and in several USA cities (Los Angeles, Washington-Baltimore, Indianapolis) at similar latitudes (Fig.4, lines around 310). The global background simply provides a baseline to show the strong anthropocentric emission impacts on the local $CH_4$ concentration variation.

9. Figure 1: What does Emissions (ht) mean? What unit is this?

Response: Sorry for the misleading, it is hundred ton, just like the commonly used kt representing thousand tons. We added explanations in the figure caption.

10. In terms of data from TCCON, it is well known that they do not operate well under high aerosol or cloud conditions. This region is known to have high AOD. What percentage of the total observations did not retrieve CH4 due to these conditions? Are there differences in the surface CH4 between when retrievals were successful and not? (Tu et al., 2020; Laughner et al., 2024)?

Response: The Xianghe and Hefei sites have percentages of 32.6% and 87.1% in 2019, respectively, for high AOD or cloud/rain that do not retrieve $CH_4$. With high aerosol or cloud/rain conditions, the DC signal of the interferogram has a large variation. All the spectra with the DC variation larger than 5% are filtered out, which guarantee the data quality of the TCCON spectra. In addition, the TCCON uses the solar direct absorption spectra which has a much less impact from the low aerosol as compared to the satellite retrieval (Added in lines around 220).

Indeed, the surface $CH_4$ concentrations at XH site observed by Picarro showed differences when retrievals were successful and not for TCOON (Fig.S14). And when retrievals were not successful, surface $CH_4$ usually reached peaks.

11. Why was TROPOMI data used with QA of 0.5? How many days of missing data are there when aggregating the data in time from day-to-day to your output frequency? How much missing data is there at the three spatial scales you are aggregating to (0.1x0.1? At 0.3x0.3?

at 0.5x0.5)? Are there differences in surface concentration between days with TROPOMI data and without TROPOMI data? (Qu et al., 2021; Worden et al., 2012)

Response: The'qa_value>0.5'is recommended by official producer SRON. See the descriptions of 'qa_value' at page 32 for "Sentinel-5P-Level-2-Product-User-Manual-Methane.pdf" (Apituley et al., 2022). The days with no observations were 24.8%, 7.0% and 0.0% for TROPOMI data at 0.1, 0.25, and 2.5 degree resolutions in 2019, respectively, and we further plotted the number of observations for TROPOMI in Fig. S24 (e-j), and the surface concentration between days with TROPOMI data and without TROPOMI data were different too.

12. Since you have not used the same retrieval assumptions for TCCON and for TROPOMI, therefore you cannot quantify the accuracy. Please re-word this sentence. Precision can possibly be determined, but would require an explicit treatment of the uncertainties. This work may be heavy, so please at least just talk about how you would do this in the future. (Povey et al., 2015)

Response: We deleted the word "higher-precision", and we added discussions on factors influencing the accuracy and uncertainty in lines 430. Furthermore, although the inversion algorithms and related processing methods of TCCON and TROPOMI are different, TCCON, as a calibration standard for docking with WMO, can be used as satellite authenticity test data, and the comparison between TCCON and TPOMI can be used to transmit WMO standards to satellite observations.

13. Your a priori data is annual. How is this distributed for both coal mine emissions and agricultural emissions, which are variable throughout different times of the year?

Response: We added descriptions in lines around 271. We did use annual coal emissions (GFEI v1.0), and annual livestock emissions (annual EDGAR v4.3.2), but used monthly rice emissions (annual EDGAR v4.3.2 with seasonal scaling factors). For coal emissions, emission flux is invariant from month to month; for rice emissions, the seasonality over Northern China is shown below:

[Figure]

[Figure]

14. Since the observations are daily, I would want to see spatial maps of the daily comparisons between TROPOMI and the modeled results. Especially on some highly emitting days. Other studies have already been doing this level of comparison. It could be done by using the data to filter high emissions days, or even some objective approach such as EOF. In the event of a high emissions perturbation, the transport time across the domain is more than 1 day, allowing the emissions to still be within the observed region, while at monthly scale they will have smeared around much of the northern hemisphere. (Lalongo et al., 2020; Verhoelst et al., 2021; Wang et al., 2021)

Response: We selected three high concentration (emitting) days in 2019 (Fig. S23). Although there were system errors between TROPOMI and Westlake modeled results, they showed generally consistent spatial patterns, with much higher concentrations in southern Shanxi and Henan, and lower concentration in northern Shanxi and Hebei. And the R even reached 0.6-0.7 for two cases. We added discussions and references in lines around 555.

15. Based on figure 2, the variation seems to be at daily or higher frequency, consistent with. Your analysis mentions September and January as being the most extreme, yet I cannot see it from the data. From my eyes, November, December, February, and other months also have very high values. The patterns seem more complex than seasonal.

Response: We revised accordingly in lines around 285 and FigS5. We plotted the frequency of hourly value higher than 2500 ppb in each month (Fig. S), and autumn and winter months had higher frequencies than spring and summer.

16. Figure 3: hour-by-hour observations require meteorology to interpret. Were there changes in the wind speed or boundary layer height?

Response: We plotted modeled hourly PBLH at BJ, XH, and SDZ (Fig. S7) and also hourly mean concentration at local time (Fig. 4). $CH_4$ concentrations were negatively correlated with PBLH. We added descriptions in lines around 342.

17. In Figure 3, what happened to December (D)?

Response: Thank you. We added December (D) in Figure 3.

18. Given the strong global gradient at different latitudes, perhaps comparisons should be made with other large urban areas and industrial areas at a similar latitude?

Response: Thank you. We added $CH_4$ observations at Taiyuan (Lat=37.7), Washington (Lat=38.9), Indianapolis (Lat=39.6), and Los Angeles (Lat=34.1) in Fig.4. And we added comparison in lines around 340.

19. Can Figure 4 include comparisons at daily scale?

Response: We added daily comparisons in Fig.S11-S13 and in lines around 436, and as expected the correlations become much weaker.

20. In Figure 4 TROPOMI looks very low compared to TCCON at XH, much more so than the uncertainty given in the paper. Please explain.

Response: Thank you. We further analyzed it at three resolutions at 0.1, 0.25, and 2.5 degree (Fig. S10), and the mean bias increased with the TROPOMI resolution, from -4.2 to -13.8 ppb in XH, and -0.7 to 4.4 ppb in HF, but lower resolution would match less TROPOMI observations, indicating a moderate resolution is needed (added in lines around 442).

21. In Figure 4, how to explain the cases where TROPOMI is higher than PICARRO in BJ?

Response: Sorry for the misleading. This is a two y-axis plot with different scales, left side y-axis is Picarro, and right side y-axis is TCOON (added in the figure caption). And using the same scale showed much lower TCCON values (Fig. S11).

22. Where daily comparisons are made (In Figure 7) the RMSE error ranges from 110ppb to 185ppb, which is in the range from 5% to 10% of the xCH4. If this is robust, this is an important finding for the satellite community. Please highlight this result.

Response: Thank you. We highlighted this results in the abstract. And Fig. 7 is surface comparisons of Picarro observations with Westlake model. We also plotted the $XCH_4$ comparisons of TROPOMI and Westlake model, which showed consistent patterns too (Fig. S19-S20).

23. The comparison with CAMS is only done monthly, and does not seem to offer any improvement to the story. Plus CAMS has been found to not be a good match with other remotely-sensed datasets over China. If there is no added value, perhaps this part can be left out, so the other parts can be focused on more clearly? (Liu Z. et al., 2024; Zhang et al., 2022).

Response: Thank you for this point, we revised accordingly. CAMS is a global inversion product that only provides a baseline for the regional background with a coarse resolution of 1°×1°.

24. Do the high emissions spots in Figure 9 match the known locations of high gas emitting coal mines and oil fields in Shanxi and other areas? How can the artificially low urban signals in Nanjing and other heavily urban parts of the southern edge of the domain, as well as the artificially low coal signals in southwestern Shanxi along the western edge be explained? Is this an artifact of being located near the edge of the domain? (Cohen and Prinn, 2011)

Response: As stated at the beginning, after re-drew figures, the high emissions spots in Figure 9-11 matched well with the known locations of high gas emitting coal mines and oil fields in Shanxi, Henan, and other areas (Fig. S17-S18). The urban signals in Nanjing were from medium to high, the southwestern Shanxi along the western edge also showed medium to high signals, which is consistent with the TROPOMI spatial pattern (Fig. S19), and also generally consistent with (Han et al., 2024; Peng et al., 2023), moreover, the relatively coarse model resolution reduced the details due to the average of the larger grids. And this indeed calls for finer scale inversion modelling studies (added in lines around 602). The inversion was conducted for the whole China (Liang et al., 2023), and we zoom out to see a larger region, and this was not an artifact of being located near the edge of the domain, as described in (Cohen and Prinn, 2011). However, for the northern edge of Shanxi, there were high emissions in PKU inventory, but not in the TROPOMI observations (Fig.S24), and thus no high concentrations in this region in the Westlake results were found.

References:

Butz, A., Galli, A., Hasekamp, O., Landgraf, J., Tol, P., and Aben, I.: TROPOMI aboard Sentinel-5 Precursor: Prospective performance of CH4 retrievals for aerosol and cirrus loaded atmospheres, Remote Sens. Environ., 120, 267–276, https://doi.org/10.1016/j.rse.2011.05.030, 2012.

Chen, Y.-H. and Prinn, R. G.: Atmospheric modeling of high- and low-frequency methane observations: Importance of interannually varying transport, J. Geophys. Res., 110, D10303, https://doi.org/10.1029/2004JD005542, 2005.

Cohen, J. B., & Prinn, R. G. (2011). Development of a fast, urban chemistry metamodel for inclusion in global models. Atmospheric Chemistry and Physics, 11(15), 7629–7656. https://doi.org/10.5194/acp-11-7629-2011

Guanter, L., Irakulis-Loitxate, I., Gorroño, J., Sánchez-García, E., Cusworth, D. H., Varon, D. J., Cogliati, S., & Colombo, R. (2021). Mapping methane point emissions with the PRISMA spaceborne imaging spectrometer. Remote Sensing of Environment, 265, 112671. https://doi.org/10.1016/j.rse.2021.112671

Hu, W., Qin, K., Lu, F. et al. Merging TROPOMI and eddy covariance observations to quantify 5-years of daily CH4 emissions over coal-mine dominated region. Int J Coal Sci Technol 11, 56 (2024). https://doi.org/10.1007/s40789-024-00700-1

Kang, H., Qin, K., Fan, L., Wei, H., Qing, X., and Cohen, J.B.: Methane point sources emission characteristics of coal industry in Shanxi Province based on Gaofen-5 satellite, J. China Coal Soc., https://doi.org/10.13225/j.cnki.jccs.2023.1247, 2024.

Kirschke, S., Bousquet, P., Ciais, P. et al. Three decades of global methane sources and sinks. Nature Geosci 6, 813–823 (2013). https://doi.org/10.1038/ngeo1955

Krummel, P. B., Montzka, S. A., Harth, C., Miller, B. R., Mühle, J., Dlugokencky, E. J., Salameh, P. K., Hall, B. D., O'Doherty, S., Steele, L. P., Dutton, G. S., Young, D., Nance, J. D., Elkins, J. W., Arnold, T., Miller, L., Fraser, P. J., Derek, N., Weiss, R. F., … Prinn, R. G. (n.d.). Overview of comparisons of non-CO2 trace gas measurements between AGAGE and NOAA at common sites.

Lalongo, I., Virta, H., Eskes, H., Hovila, J., & Douros, J. (2020). Comparison of TROPOMI/Sentinel-5 Precursor NO2 observations with ground-based measurements in Helsinki. Atmospheric Measurement Techniques, 13(1), 205–218. https://doi.org/10.5194/amt-13-205-2020

Laughner, J. L., Toon, G. C., Mendonca, J., Petri, C., Roche, S., Wunch, D., Blavier, J.-F., Griffith, D. W. T., Heikkinen, P., Keeling, R. F., Kiel, M., Kivi, R., Roehl, C. M., Stephens, B. B., Baier, B. C., Chen, H., Choi, Y., Deutscher, N. M., DiGangi, J. P., … Wennberg, P. O. (2024). The Total Carbon Column Observing Network's GGG2020 data version. Earth System Science Data, 16(5), 2197–2260. https://doi.org/10.5194/essd-16-2197-2024

Li, X., Cohen, J. B., Qin, K., Geng, H., Wu, X., Wu, L., Yang, C., Zhang, R., and Zhang, L.: Remotely sensed and surface measurement-derived mass-conserving inversion of daily NOx emissions and inferred combustion technologies in energy-rich northern China, Atmos. Chem. Phys., 23, 8001–8019, https://doi.org/10.5194/acp-23-8001-2023, 2023.

Liu, Y., Qin, K., Cohen, J. B., Kang, H., Hu, W., Lu, F., Wu, X., and Yang, C.: Analysis of the characteristics of methane in the coal mining area of southeastern Shanxi with eddy and mobile observation, J. China Coal Soc., 47, 4395–4402, 2022.

Liu, Z., Cohen, J. B., Wang, S., Wang, X., Tiwari, P., & Qin, K. (2024). Remotely sensed BC columns over rapidly changing Western China show significant decreases in mass and inconsistent changes in number, size, and mixing properties due to policy actions. Npj Climate and Atmospheric Science, 7(1), 1–16. https://doi.org/10.1038/s41612-024-00663-9

Pei, Z., Han, G., Mao, H., Chen, C., Shi, T., Yang, K., Ma, X., & Gong, W. (2023). Improving quantification of methane point source emissions from imaging spectroscopy. Remote Sensing of Environment, 295, 113652. https://doi.org/10.1016/j.rse.2023.113652

Plant, G., Kort, E. A., Murray, L. T., Maasakkers, J. D., and Aben, I.: Evaluating urban methane emissions from space using TROPOMI methane and carbon monoxide observations, Remote Sens. Environ., 268, 112756, https://doi.org/10.1016/j.rse.2021.112756, 2022.

Prinn, R. G., Weiss, R. F., Arduini, J., Arnold, T., DeWitt, H. L., Fraser, P. J., … Zhou, L. (2018). History of Chemically and Radiatively Important Atmospheric Gases from the

Advanced Global Atmospheric Gases Experiment (AGAGE). https://doi.org/10.5194/essd-2017-134

Qin, K., Hu, W., He, Q., Lu, F., and Cohen, J. B.: Individual coal mine methane emissions constrained by eddy covariance measurements: low bias and missing sources, Atmos. Chem. Phys., 24, 3009–3028, https://doi.org/10.5194/acp-24-3009-2024, 2024.

Qin, K., Lu, L., Liu, J., He, Q., Shi, J., Deng, W., Wang, S., and Cohen, J. B.: Model-free daily inversion of NOx emissions using TROPOMI (MCMFE-NOx) and its uncertainty: Declining regulated emissions and growth of new sources, Remote Sens. Environ., 295, 113720, https://doi.org/10.1016/j.rse.2023.113720, 2023.

Qu, Z., Jacob, D. J., Shen, L., Lu, X., Zhang, Y., Scarpelli, T. R., Nesser, H., Sulprizio, M. P., Maasakkers, J. D., Bloom, A. A., Worden, J. R., Parker, R. J., and Delgado, A. L.: Global distribution of methane emissions: a comparative inverse analysis of observations from the TROPOMI and GOSAT satellite instruments, Atmos. Chem. Phys., 21, 14159–14175, https://doi.org/10.5194/acp-21-14159-2021, 2021.

Rivera Martinez, R. A., Santaren, D., Laurent, O., Broquet, G., Cropley, F., Mallet, C., Ramonet, M., Shah, A., Rivier, L., Bouchet, C., Juery, C., Duclaux, O., & Ciais, P. (2023). Reconstruction of high-frequency methane atmospheric concentration peaks from measurements using metal oxide low-cost sensors. Atmospheric Measurement Techniques, 16(8), 2209–2235. https://doi.org/10.5194/amt-16-2209-2023

Tu, Q., Hase, F., Blumenstock, T., Kivi, R., Heikkinen, P., Sha, M. K., Raffalski, U., Landgraf, J., Lorente, A., Borsdorff, T., Chen, H., Dietrich, F., & Chen, J. (2020). Intercomparison of atmospheric CO2 and CH4 abundances on regional scales in boreal areas using Copernicus Atmosphere Monitoring Service (CAMS) analysis, COllaborative Carbon Column Observing Network (COCCON) spectrometers, and Sentinel-5 Precursor satellite observations. Atmospheric Measurement Techniques, 13(9), 4751–4771. https://doi.org/10.5194/amt-13-4751-2020

Verhoelst, T., Compernolle, S., Pinardi, G., Lambert, J.-C., Eskes, H. J., Eichmann, K.-U., Fjæraa, A. M., Granville, J., Niemeijer, S., Cede, A., Tiefengraber, M., Hendrick, F., Pazmiño, A., Bais, A., Bazureau, A., Boersma, K. F., Bognar, K., Dehn, A., Donner, S., … Zehner, C. (2021). Ground-based validation of the Copernicus Sentinel-5P TROPOMI NO2 measurements with the NDACC ZSL-DOAS, MAX-DOAS and Pandonia global networks. Atmospheric Measurement Techniques, 14(1), 481–510. https://doi.org/10.5194/amt-14-481-2021

Wang, S., Cohen, J. B., Deng, W., Qin, K., & Guo, J. (2021). Using a New Top-Down Constrained Emissions Inventory to Attribute the Previously Unknown Source of Extreme Aerosol Loadings Observed Annually in the Monsoon Asia Free Troposphere. Earth's Future, 9(7). https://doi.org/10.1029/2021EF002167

Povey, A. C. and Grainger, R. G.: Known and unknown unknowns: uncertainty estimation in satellite remote sensing, Atmos. Meas. Tech., 8, 4699–4718, https://doi.org/10.5194/amt-8-4699-2015, 2015.

Worden, J., Kulawik, S., Frankenberg, C., Payne, V., Bowman, K., Cady-Peirara, K., Wecht, K., Lee, J.-E., and Noone, D.: Profiles of CH4, HDO, H2O, and N2O with improved lower tropospheric vertical resolution from Aura TES radiances, Atmos. Meas. Tech., 5, 397–411, https://doi.org/10.5194/amt-5-397-2012, 2012.

Zhang, B., Chellman, N. J., Kaplan, J. O., Mickley, L. J., Ito, T., Wang, X., Wensman, S. M., McCrimmon, D., Steffensen, J. P., McConnell, J. R., & Liu, P. (2024). Improved biomass burning emissions from 1750 to 2010 using ice core records and inverse modeling. Nature Communications, 15(1), 3651. https://doi.org/10.1038/s41467-024-47864-7

Zhang, Y., Li, J., Li, J., Pan, X., Wang, W., Zhu, L., Wang, Z., Chen, X., Yang, W., & Wang, Z. (2022). An intercomparison of ozone taken from the Copernicus atmosphere monitoring service and the second Modern-Era retrospective analysis for research and applications over China during 2018 and 2019. Journal of Environmental Sciences, 114, 514–525. https://doi.org/10.1016/j.jes.2022.01.045

Apituley, A., Pedergnana, M., Sneep, M., et al., 2022. Sentinel-5 precursor/TROPOMILevel 2 Product User ManualMethane. Available at https://sentinel.esa.int/documents/247904/2474726/Sentinel-5P-Level-2-Product-User-Manual-Methane.pdf/1808f165-0486-4840-ac1d-06194238fa96, accessed on 30 October 2024.

Bao, Z., Han, P., Zeng, N., et al., 2020. Observation and modeling of vertical carbon dioxide distribution in a heavily polluted suburban environment. Atmospheric and Oceanic Science Letters, 1-9.

Han, G., Pei, Z., Shi, T., et al., 2024. Unveiling Unprecedented Methane Hotspots in China's Leading Coal Production Hub: A Satellite Mapping Revelation. Geophysical Research Letters 51 (10), e2024GL109065.

Hu, C., Xiao, W., Griffis, T.J., et al., 2023. Estimation of Anthropogenic CH4 and CO2 Emissions in Taiyuan-Jinzhong Region: One of the World's Largest Emission Hotspots. Journal of Geophysical Research: Atmospheres 128 (8), e2022JD037915.

Liang, R., Zhang, Y., Chen, W., et al., 2023. East Asian methane emissions inferred from high-resolution inversions of GOSAT and TROPOMI observations: a comparative and evaluative analysis. Atmos. Chem. Phys. 23 (14), 8039-8057.

Peng, S., Giron, C., Liu, G., et al., 2023. High-resolution assessment of coal mining methane emissions by satellite in Shanxi, China. iScience 26 (12), 108375.

Yu, X., Millet, D.B. and Henze, D.K., 2021. How well can inverse analyses of high-resolution satellite data resolve heterogeneous methane fluxes? Observing system simulation experiments with the GEOS-Chem adjoint model (v35). Geosci. Model Dev. 14 (12), 7775-7793.

Zeng, N., Han, P., Liu, Z., et al., 2021. Global to local impacts on atmospheric CO2 from the COVID-19 lockdown, biosphere and weather variabilities. Environmental Research Letters 17 (1), 015003.

Zhang, B., Chellman, N.J., Kaplan, J.O., et al., 2024. Improved biomass burning emissions from 1750 to 2010 using ice core records and inverse modeling. Nature Communications 15 (1), 3651.

**Reviewer #2**

Han et al. present an evaluation of various methane measurements from northern China, analyzing the spatiotemporal variability of concentration data from both measurements and models for the 2019-2021 period. In the current era, there are many different kinds of methane observations, and it is not always clear how to effectively use the combined strengths of these datasets. In that sense, this study undertakes an important task by attempting to cross-validate and reconsider the variability in different types of measurements. The region analyzed are also very important, as it includes one of the largest anthropogenic emissions hotspots.

However, the study has some major shortcomings, which is why I am recommending major revisions. The primary issue is the exclusive focus on concentration data. The ultimate goal should be to understand methane emissions (and sinks). Concentration data can be used as a proxy for emissions, but attempts must be made to convert concentrations into quantities that more accurately reflect emissions variability (spatial or temporal). Simply comparing concentration data from different sources is insufficient, especially when the data represent different types of measurements (e.g., in-situ vs. total column).

Response: We thank you for your understanding of this paper and all the good suggestions. We agree with you that the ultimate goal is to understand methane emissions and sinks. And the concentrations data are the foundations for atmospheric inversions, and thus the main contribution of this paper mainly focuses on concentration and enhancement analyses, and provided high-precision datasets to the modelling community for further inversion studies (Han et al., 2024; Mitchell et al., 2022; Yang et al., 2021). The Westlake model results were from a East Asia inversion using TROPOMI observation data (Liang et al., 2023), we added more analyses on CH4 enhancement and emissions (Fig. 5, 11, 14, S17-18, S25), CH4 prior and posterior (Fig.14), and the fully local inversions are beyond the scope of this study, the high-resolution inversion runs are still in progress on our server, and will be presented in another separate paper.

**Major Comments:**

**Emission Representation from Concentration Data**: I would like to see a more robust attempt to convert concentration data into a metric that represents emissions.

This can be achieved in various ways, but at the very least, a local/regional background should be subtracted from the concentration data. While defining the correct background is challenging, such an attempt would be valuable and may improve the correlation values presented in the paper.

Response: Thank you. We selected SDZ (Shangdianzi), a WMO/GAW regional background station as the study regional background, as stated in lines around 154, and following your suggestions, we analyzed enhancement and emissions in Fig.5, and surface concentrations with PKU-v2 emissions (Fig.14, S17-18, S25), to check the correlations of enhancements, concentrations with emissions. The results showed that correlations of $R^2$ increased to 0.2~0.3 for surface concentrations with emissions.

1. **Model-Derived Correlations**: Using models like CAMS or GEOS-Chem, it should be possible to estimate the expected correlation between different measurement types. For example, I would be interested to see what correlation would emerge from CAMS/GEOS-Chem-simulated observations between TCCON, TROPOMI, in-situ data, and emissions. This would provide a reference for interpreting the correlation analyses and contextualize the low correlation values presented in the study.

Response: Thank you, and we added all these analyses from Fig.S9, Fig.S15-16, S19, S20, S23, and compared the GEOS-Chem simulated observations with ground-based and TROPOMI data. Model captured the variations of in situ observations (Fig. 10; S15-S16), and the trends generally well with TCOON site observations (although with systematic biases, Fig. S9), and also the seasonal and daily spatial patterns with TROPOMI (Fig. S19-S20, S23). The errors were larger in BJ than in SDZ, indicating a coarse resolution of ~50km could not capture all the in-situ urban signals well. And comparisons with PKU-v2 emissions showed consistent spatial patterns (Fig, S17-S18) and correlations (Fig. 14, S24).

2. **Inversion Emissions Estimates**: It is unfortunate that the paper offers only a very limited analysis of inventory and inversion emissions estimates. These estimates are among the closest to true emissions of all the datasets presented, and I believe there should be a more substantial investigation of the inventory and GEOS-Chem posterior emissions.

Response: We thank you for this point. This paper mainly focuses on concentration and enhancement analyses, and provided high-precision datasets to the modelling community for further fine scale high-resolution inversion studies, as we have done for the CO2 datasets (Yang et al., 2021), and thus fully local inversions are beyond the scope of this study, and the high-resolution inversion runs are still in progress on our server, and will be presented in another separate paper. However, to incorporate your good suggestions, we added more analyses on regional inversion and inventories (PKU-v2 and EDGAR-v4.3.2) results (added in Fig. 15; section 3.5, lines around 565)

and concentration gradients associated with emissions (Fig. 5). We also added a literature review on $CH_4$ emissions of this region (Table S2).

Directly related to the emission of a region is the concentration gradient. Therefore, the concentration gradient difference between XH-SDZ and BJ-SDZ were discussed using SDZ as the background station, and the concentration gradient difference were consistent with emissions enhancement (Fig. 5b-c).

Moreover, Westlake model simulation (with a 50 km resolution), can catch the concentration difference of XH-SDZ, but not the concentration difference of BJ-SDZ. And thus it shows that the analyses based on satellite observations (TROPOMI) as model input and inversions (Westlake product) are not very effective in resolving the distribution of emissions within the city, and also shows that it is necessary to use BJ, XH, SDZ observational data in higher resolution models and inversion systems. We added these discussions in lines around 466 and 574.

3. **Column vs. In-situ Concentration Observations**: The authors need to first clarify why these different measurements can be directly compared. Column and in-situ observations represent different quantities, and in-situ data, for instance, is strongly influenced by boundary layer height variability.

Response: We added clarification for the comparison between column vs. in-situ concentration observations for our data analyses and reference (Ialongo et al., 2020) (lines 382, Fig.S22). There are indeed caveats in comparing the two measurements. Since TROPOMI only provide $XCH_4$ (no surface concentrations) and most ground-based measurements are in-situ, these two measurements can be combined to expand the data set. For example, we found good correlations between $XCO_2$ and in-situ $CO_2$ at Xianghe and Shangdiani stations (Han et al., 2024), but not in urban Beijing, where signals are more influenced by local emissions. Similar trends were found in $CH_4$ comparisons (Fig.8 a-b, Fig. S11-S13, lines around 405).

4. **Line 312**: A strong correlation is not expected as TROPOMI averages over a larger horizontal spatial area, which is further diluted by column averaging. However, TROPOMI also senses local/regional emissions. A deeper investigation into the very poor correlation needs to be conducted. I wonder if the correlation improves when different spatial and/or temporal averaging methods are applied. Also, how much of the variability of TROPOMI data is dictated by data coverage gaps needs to be discussed.

Response: We further analyzed three spatial resolutions (0.1, 0.5, and 2.5 degrees) for TROPOMI to compare with in-situ measurements (Fig. S10). The results showed that biases between TROPOMI and TCOON increased with resolution from 0.1 to 2.5 for XH and HF (added in lines around 412), and the data gap indeed influenced very much (Fig. S23-S24), e.g. in the northern Shanxi, the data gap was more serious, and

thus the posterior concentration results were not very consistent with emissions (Fig. S24 d,e,f).

**Minor Comments:**

• **Figures**: Panels within figures need to be consolidated. For example, panels in Figure 6 are on separate pages, and the same issue occurs with Figure 9.

Response: Thank you. We revised Figure 6 and 9 accordingly.

• **Line 94**: "due mainly to" should be revised to "mainly due to."

Response: Thank you. We revised.

• **Line 100**: "measurements of trends" should be revised to "measurements."

Response: We deleted accordingly.

• **Line 130**: "quantify the spatiotemporal CH4 concentrations" should be revised to "quantify the spatiotemporal variations of CH4 concentrations."

Response: We added "variations of".

• **Line 245**: This sentence is overly complex. Consider rephrasing for clarity.

Response:    Thank you for pointing out. We rephrased the long sentences into short ones, and revised in lines around 260.

• **Line 264**: "To determine the importance of understanding the uncertainties in concentrations…" should be revised to "To understand the errors in concentrations…"

Response: We revised accordingly.

• **Figure 9 (Annual Panel)**: Correct "XH4" to "XCH4" in the figure legends.

Response: Thank you. Revised accordingly.

• **Figure Captions**: More detailed information is needed. For instance, the caption for Figure 1 is too brief considering the amount of data it presents.

Response: Thank you. We added more information for the figure captions.

Han, P., Yao, B., Cai, Q., et al., 2024. Support Carbon Neutral Goal with a High-resolution Carbon Monitoring System in Beijing, accepted by BAMS. Doi: 10.1175/BAMS-D-23-0025.1. Available at https://journals.ametsoc.org/view/journals/bams/aop/BAMS-D-23-0025.1/BAMS-D-23-0025.1.xml, accessed on November 15th, 2024. Bulletin of the American Meteorological Society.

Ialongo, I., Virta, H., Eskes, H., Hovila, J. and Douros, J., 2020. Comparison of TROPOMI/Sentinel-5 Precursor NO2 observations with ground-based measurements in Helsinki. Atmos. Meas. Tech. 13 (1), 205-218.

Liang, R., Zhang, Y., Chen, W., et al., 2023. East Asian methane emissions inferred from high-resolution inversions of GOSAT and TROPOMI observations: a comparative and evaluative analysis. Atmos. Chem. Phys. 23 (14), 8039-8057.

Mitchell, L.E., Lin, J.C., Hutyra, L.R., et al., 2022. A multi-city urban atmospheric greenhouse gas measurement data synthesis. Scientific Data 9 (1), 361.

Yang, Y., Zhou, M., Wang, T., et al., 2021. Spatial and temporal variations of CO2 mole fractions observed at Beijing, Xianghe, and Xinglong in North China. Atmos. Chem. Phys. 21 (15), 11741-11757.

---

## Author Response (AR2)

**Response to reviewer#1**

A few technical correlations:

line 227: "7 km×7 km before August 2019" is better. Operational TROPOMI measurements began in May 2018.
Response: Thank you, we revised accordingly.

Line 347: Either remove or provide further clarification for this sentence: "Moreover, there is a very small fraction of negative values for enhancement calculated based on SDZ as background, indicating a not real background for all climatic conditions, and this message is important in atmospheric inversions." To me, occasional negative enhancements for a small temporal grid are not very surprising, as even the background sites can have random analytical errors and be affected by emissions.
Response: Thank you, we removed this sentence.

Line 410: "TPOMI"=> "TROPOMI"
Response: Thank you, we revised accordingly.

Line 453: "introduced wrong message in both" => "introduced errors in both"
Response: Thank you, we revised accordingly.

Line 493: "system errors": Do the authors mean "systematic errors"?
Response: Yes, we mean "systematic errors" and revised.

Supplementary Information: The figures and their captions are often on different pages. Please fix this formatting issue.
Response: Thank you, we revised accordingly.

**Response to reviewer#2**

Dear Editor and Authors:

Overall, the authors have improved their work greatly as compared with before. Their analysis is deeper. They have referenced more papers. They have started to work with higher frequency data in space and time. They have talked more about strengths and

weaknesses of existing platforms. They have provided access to their dataset, which will allow the community to continue to grow and improve.

In addition to all of this, the authors mention an important caveat of their paper in their response, and I hope that they place it clearly in the paper's final version. The meaning is that the surface data provides a more accurate representation of higher CH4 loadings than observed by TCOON, and even more so than from TROPOMI. While they have not deeply performed this analysis of how much an impact of a low bias solely relying on low-resolution (space and time) models and observations may have, they clearly demonstrate this. Placing some comments about this weakness or in the direction of future research will add to the value of the paper, since it is via high frequency surface measurements like those provided, that such future improvements by the community can be made.

"Indeed, the surface CH4 concentrations at XH site observed by Picarro showed differences when retrievals were successful and not for TCOON (Fig.S14). And when retrievals were not successful, surface CH4 usually reached peaks."

Response: Thank you for this good point, and we added this important caveat in lines 381-383.

One final issue I wanted to raise is my question about elevation was not regarding the model, but instead how the observations were calibrated with respect to the reduced air pressure found in Shanxi and other locations in China's center and west, which are at a higher elevation than those from the TCCON stations. I wanted to check if such a concentration calibration correction was applied, and if so, how well it performed.

Response: Thank you for this question, and we added the discussions and short comings for this question in lines 384-389. The reduced air pressure were not considered in the comparisons across different sites (Beijing, Xianghe, Shangdianzi, and Hefei), and the elevations of these four sites were less than 500m, which is within the boundary layer and mixed well for $CH_4$ concentrations. But we acknowledge that it is better to consider this factor when comparing with the higher elevation observations such as in Shanxi sites (e.g. higher than 2 km).

I strongly support the current version's publication, once a few minor spelling, grammar, and organizational issues are clarified, and this caveat that the authors point out their data can offer an improvement to from the current community approaches, is more clearly written into the conclusions/findings. I am certain that it will make an important and positive contribution to the understanding of methane concentrations and their changes in China.

Response: We check the MS thoroughly and revised small errors (e.g. lines 416, 457, 496). We thank you for the understanding of this paper and for the careful review, which substantially improved the quality of this paper.